# Unequal mitochondrial segregation promotes asymmetric fates during neurogenesis

Benjamin Bunel [1,2], Rémi Leclercq[1], Rosette Goïame[1], Arnaud Gautier [3,4], Xavier Morin [1,5] ✉ & Evelyne Fischer [1,5] ✉

Asymmetric cell division plays a critical role during vertebrate neurogenesis by generating neuronal cells while maintaining a pool of progenitors. It relies on unequal distribution of cell fate determinants during progenitor division. Here, we use live imaging in the chick embryonic neuroepithelium to demonstrate that mitochondria behave as asymmetric fate determinants during mitosis. We show that the frequency of unequal distribution of mitochondria increases in parallel with the rate of asymmetric divisions during development. Furthermore, fate tracking experiments reveals that following progenitor division, a cell inheriting fewer mitochondria than its sister consistently differentiates into a neuron. We set up a chemogenetic approach to experimentally displace mitochondria specifically during mitosis to force their unequal inheritance and find that this drives premature neuronal differentiation. In this work, we establish a direct causal relationship between unequal mitochondrial inheritance and the asymmetric fate of sister cells in vivo, revealing a pivotal mechanism for neurogenesis.

Asymmetric division is a fundamental mechanism used by stem cells to drive commitment of one daughter cell to differentiation while simultaneously maintaining stemness in the other daughter cell. Mechanistically, asymmetric cell division relies on the unequal segregation during mitosis of fate determinants of various molecular nature (such as mRNAs, proteins, organelles and vesicles) present in the mother cell, which initiate the establishment of different cellular programs between daughter cells. Although numerous fate determinants have been identified in invertebrate model organisms, only a few have been characterized in vertebrates[1], in part due to the difficulty of monitoring cell division events and subsequent daughter cells fate acquisition in vivo.

Mitochondria have recently emerged as essential players in fate transitions in several models. Following pioneer studies in yeast[2], recent work in *C. elegans*[3] and in vertebrate stem cell models in vitro

has correlated the unequal segregation of the mother cell's mitochondrial pool during mitosis with differential fate in their progeny[4–8]. However, whether the differential distribution of mitochondria during mitosis is directly responsible for cell fate determination remains to be determined.

During vertebrate central nervous system development, the process of neuronal differentiation requires remodeling of mitochondrial functions and metabolism that control downstream gene regulatory pathways[9,10]. Notably, fate decisions in the progeny of cortical progenitors involve modifications in the balance between mitochondrial fusion and fission taking place within a brief time window following cell division[10]. Yet again, the question remains of whether an asymmetric mitochondrial behavior within the dividing mother cell triggers differential fate decisions in the progeny during vertebrate neurogenesis.

[1]Institut de Biologie de l'Ecole Normale Supérieure (IBENS), CNRS, Inserm, Ecole Normale Supérieure, PSL Research University, Paris, France. [2]Sorbonne Université, Collège Doctoral, Paris, France. [3]Sorbonne Université, École Normale Supérieure, Université PSL, CNRS, Laboratoire des Biomolécules, Paris, France. [4]Institut Universitaire de France, Paris, France. [5]These authors contributed equally: Xavier Morin, Evelyne Fischer.
✉e-mail: xavier.morin@ens.fr; evelyne.fischer@ens.fr

Here, we monitored mitochondrial segregation upon the division of progenitors at different stages of neurogenesis and in conditions that either promote or reduce asymmetric divisions in the chick embryonic neuroepithelium. We observed unequal segregation of the mitochondrial pool at a frequency that mirrors the rate of asymmetric divisions in these different conditions. Long-term fate tracking revealed that a cell inheriting fewer mitochondria than its sibling consistently becomes a neuron. Finally, we show that forcing unequal inheritance drives premature neuronal differentiation. Our study establishes a causal relationship between unequal mitochondrial inheritance and the asymmetric fate of sister cells in vivo.

## Results

### Uneven mitochondrial segregation during neurogenesis

We studied mitochondrial segregation during neural progenitor divisions using live imaging in the embryonic chick neural tube. Constructs expressing fluorescent proteins fused to the mitochondrial targeting sequence of Cox8 were electroporated in ovo (Fig. 1a) and mitochondrial inheritance was monitored ex ovo in "en-face" live imaging of the spinal neuroepithelium. This provides optimal access to its apical surface (Fig. 1b), where divisions occur. Using 4D confocal microscopy, we monitored mitochondria and cell contours through the whole mitotic sequence at 3 min time intervals (Fig. 1b, Supplementary Fig. 1a and Supplementary Movie 1). The mitochondrial volume inherited by each of the two sister cells immediately after cytokinesis was reconstructed and measured. We used a metric termed $R_{mito}$, which represents the ratio of the smaller to the larger of these two measurements, in order to compare mitochondrial inheritance between sister cells (Supplementary Fig. 1b, Supplementary Movie 2). A $R_{mito}$ value of 1 indicates that each daughter cell inherits a similar mitochondrial volume (50% of the mother's mitochondrial pool), whereas a $R_{mito}$ value of 0.5 indicates that one cell inherits twice as many mitochondria as its sister (Fig. 1c, Supplementary Fig. 1b). We first monitored mitochondrial inheritance at embryonic day 3 (E3, HH17-18), and found that $R_{mito}$ distributed across a broad range of values between 1 and 0.5 (Fig. 1d). At that stage, three modes of progenitor divisions coexist in the spinal cord: symmetric proliferative divisions (producing two progenitors: PP), asymmetric neurogenic divisions (producing a progenitor and a neuron: PN), and a small proportion of symmetric terminal neurogenic divisions (producing two neurons: NN)[11,12].

We wondered whether the broad distribution observed at E3 might be linked to the coexistence of these three modes of division. To test this hypothesis, we monitored mitochondrial inheritance at E2, when the symmetric PP mode of division strongly predominates[11,12]. We found that mitochondria were mostly distributed evenly between sister cells, contrasting with the high frequency of uneven inheritance observed at E3 (Fig. 1d). We then took advantage of *Tis21*, a gene activated during the transition to neurogenic divisions[13], to label a subpopulation of progenitors with a higher content in asymmetric PN divisions. We have previously shown using a reporter system based on the targeted insertion of Cre recombinase at the *Tis21* locus that the proportion of PN is doubled in the subpopulation of Tis21-expressing progenitors compared to the overall progenitor population at E3[14]. Using the same approach to activate a Cre-recombinase-dependent mitochondrial reporter (Fig. 1d and "Methods"), we found that the distribution of $R_{mito}$ values shifted significantly towards more unbalanced values in this subpopulation compared to the whole progenitor population at E3. Overall, these experiments show a parallel between the developmental increase in the rate of asymmetric divisions and the frequency of unequal distribution of mitochondria.

To explore whether unequal mitochondrial volume inheritance is a consequence of unequal cell volume partitioning, we next monitored cellular volumes obtained by 3D reconstruction of cell contours in the daughter cells of Tis21-Cre progenitors after cytokinesis. We calculated a $R_{cell}$ metric representing the ratio of the volume of the cell containing the smallest mitochondrial pool to the volume of its sister cell containing the largest mitochondrial pool (Supplementary Fig. 2). In contrast to the high frequency of uneven mitochondrial inheritance, the cellular volume was nearly identical between sister cells, as illustrated by a $R_{cell}$ consistently close to 1 (Supplementary Fig. 2). The absence of correlation between $R_{mito}$ and $R_{cell}$ (Supplementary Fig. 2) indicates that unequal mitochondrial volume inheritance constitutes an intrinsic factor and is not the consequence of a differential cell volume.

### Uneven distribution of mitochondria upon changes in the rate of asymmetric divisions

Since the frequency of uneven mitochondrial inheritance mirrors the proportion of asymmetric modes of division (Fig. 1d), we investigated whether experimentally modifying the proportion of the different modes of progenitor division would impact the distribution of $R_{mito}$.

To increase the proportion of asymmetric divisions at early stages of embryonic development, we overexpressed the Cdc25b phosphatase, which has been shown to accelerate neurogenesis by promoting neurogenic modes of division[15]. Electroporation of a Cdc25b expression vector at E1.5 led to an increased production of neurons (Supplementary Fig. 3), consistent with previous results at a later stage[15]. We next measured the distribution of $R_{mito}$ in dividing progenitors, and observed a significant shift towards uneven mitochondrial inheritance: at E2.25, the distribution of $R_{mito}$ in Cdc25b overexpressing cells was significantly different from the control at the same stage (Fig. 2a), and closely resembled what is normally observed at E3 (see Fig. 1d).

Conversely, to decrease the proportion of asymmetric divisions at E3, we knocked-down *Cdkn1c/p57^{kip2}*, as we recently showed that this favors the symmetric proliferative mode of division and consequently delays neurogenesis[16]. Downregulation of *Cdkn1c* significantly reduced the proportion of uneven mitochondrial inheritance at E3 (Fig. 2b) compared to a control shRNA, and led to a distribution of $R_{mito}$ that closely matched what is normally observed at E2.25 (Fig. 2a).

We conclude from these functional experiments that the developmental increase in the frequency of unequal mitochondrial inheritance is a consequence of the appearance of asymmetric modes of division in the progenitor population.

### Inheriting the smaller pool correlates with neural differentiation

To investigate whether uneven partitioning of mitochondria is directly associated with an asymmetric cell fate, we next monitored mitochondrial inheritance in individual progenitors and tracked fate acquisition in their daughter cells through additional imaging of their apical surface for 20 to 30 h. In the developing spinal cord, which relies on direct neurogenesis, the progressive reduction of the apical footprint followed by delamination from the apical surface characterizes a differentiating neuron (N), whereas a new division event identifies a daughter cell as a progenitor (P) [ref. 17–19, and see "Methods"; Fig. 2c, d, Supplementary Fig. 4 and Supplementary Movies 3–5].

Our results from 58 pairs of sister cells show that the distribution of $R_{mito}$ differs between symmetrically and asymmetrically dividing progenitors (Fig. 2e). In the symmetrically dividing population (PP and NN divisions, $n = 24$ pairs), the values of $R_{mito}$ were homogenously distributed in a range close to one, with two exceptions. We therefore propose a threshold at 0.85 that will be used henceforth to define "equal" versus "unequal" mitochondrial inheritance (see "Methods"). In stark contrast to symmetric divisions, the values of $R_{mito}$ in asymmetric divisions were distributed between 0.55 and 1, with the majority of PN daughters ($n = 23/34$) receiving unequal mitochondrial pools ($R_{mito} < 0.85$) (Fig. 2e). This live analysis directly links mitotic events in the progenitor to cellular identity post mitosis in its progeny and

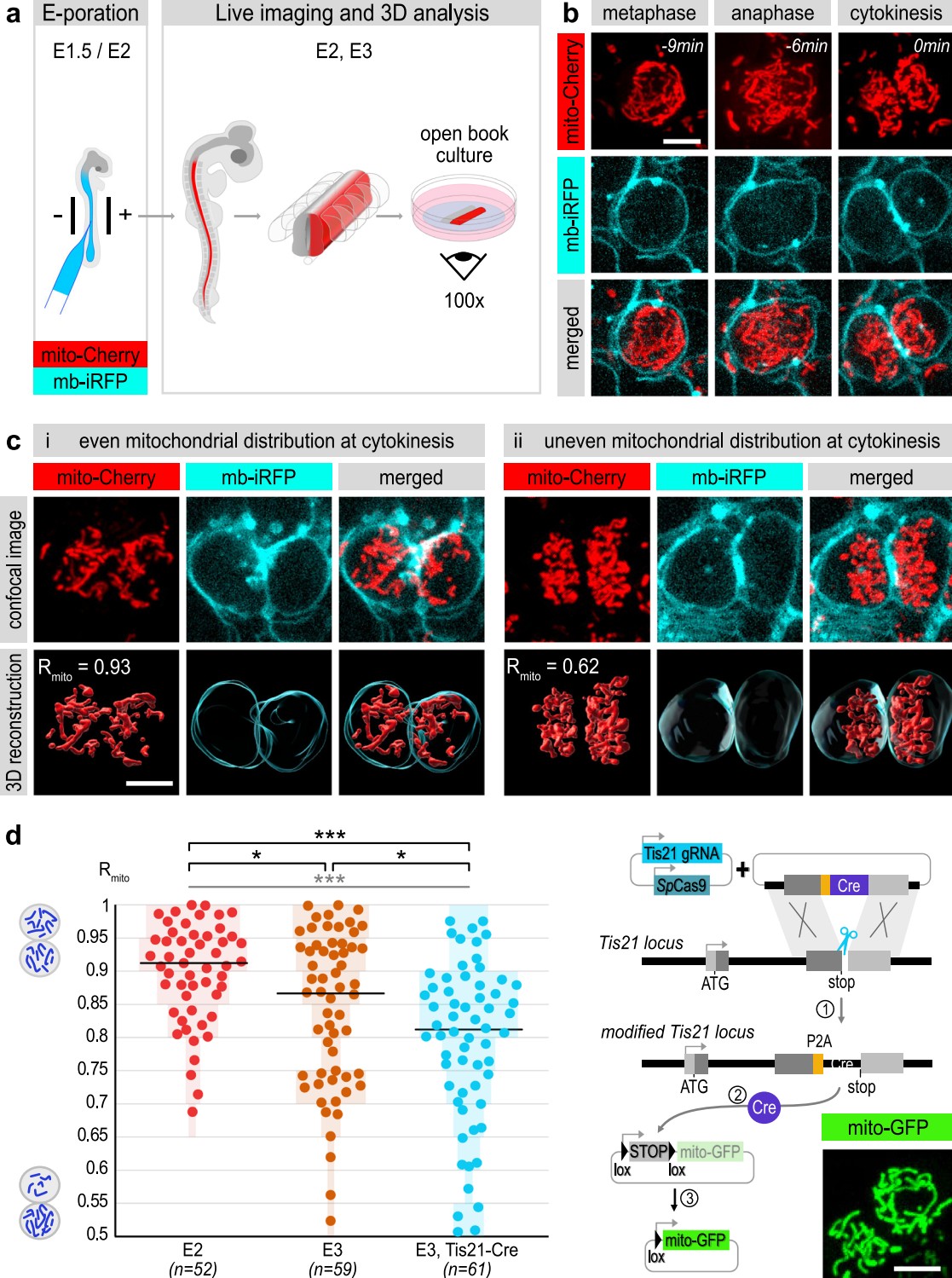

**Fig. 1 | Increased frequency of uneven mitochondrial inheritance at the onset of asymmetric modes of divisions during neurogenesis. a** Scheme of in ovo electroporation of DNA constructs in the chick embryonic neural tube and an experimental strategy for open-book live imaging in the electroporated neural tube. **b** Enface time-lapse monitoring of mitochondria (red) and cell contours (cyan) in a dividing neural progenitor. Scale bar 5 μm. **c** En-face confocal view and 3D reconstructions of mitochondrial volume in sister cells immediately after cytokinesis. 3D reconstructed images from the entire z-stacks are used to calculate the ratio $R_{mito}$ of mitochondrial inheritance between sister cells. Typical examples of even (i) and uneven (ii) inheritance are depicted. Scale bar 5 μm. **d** Left: scatter plots of $R_{mito}$ distribution at E2, E3 and in Tis21 positive progenitors (E3, Tis21-Cre). Light color

bars represent the percentage of cell pairs in each 0.05 interval. Statistical analysis: Kruskal-Wallis test ***: $p = 0.0003$ (distributions between E2, E3 and E3-Tis21-Cre), two-tailed Mann-Whitney test ***: $p = 0.001$. *: $p = 0.026$ (E2 vs E3) and *: $p = 0.015$ (E3 vs E3-Tis21-Cre). $n = 4$ embryos at E2, $n = 6$ embryos at E3 and $n = 7$ embryos for the Tis-Cre progenitors. Right: schematic representation of the CRISPR/Cas9 strategy of Cre recombinase knock-in used to activate a loxP-dependent mitochondrial reporter specifically in Tis21-Cre neurogenic progenitors. The inset shows an example of Tis-Cre driven mito-GFP in a mitotic progenitor in an en-face confocal view. Scale bar 5 μm. E = Embryonic day; min = minutes. Source data are provided as a Source Data file.

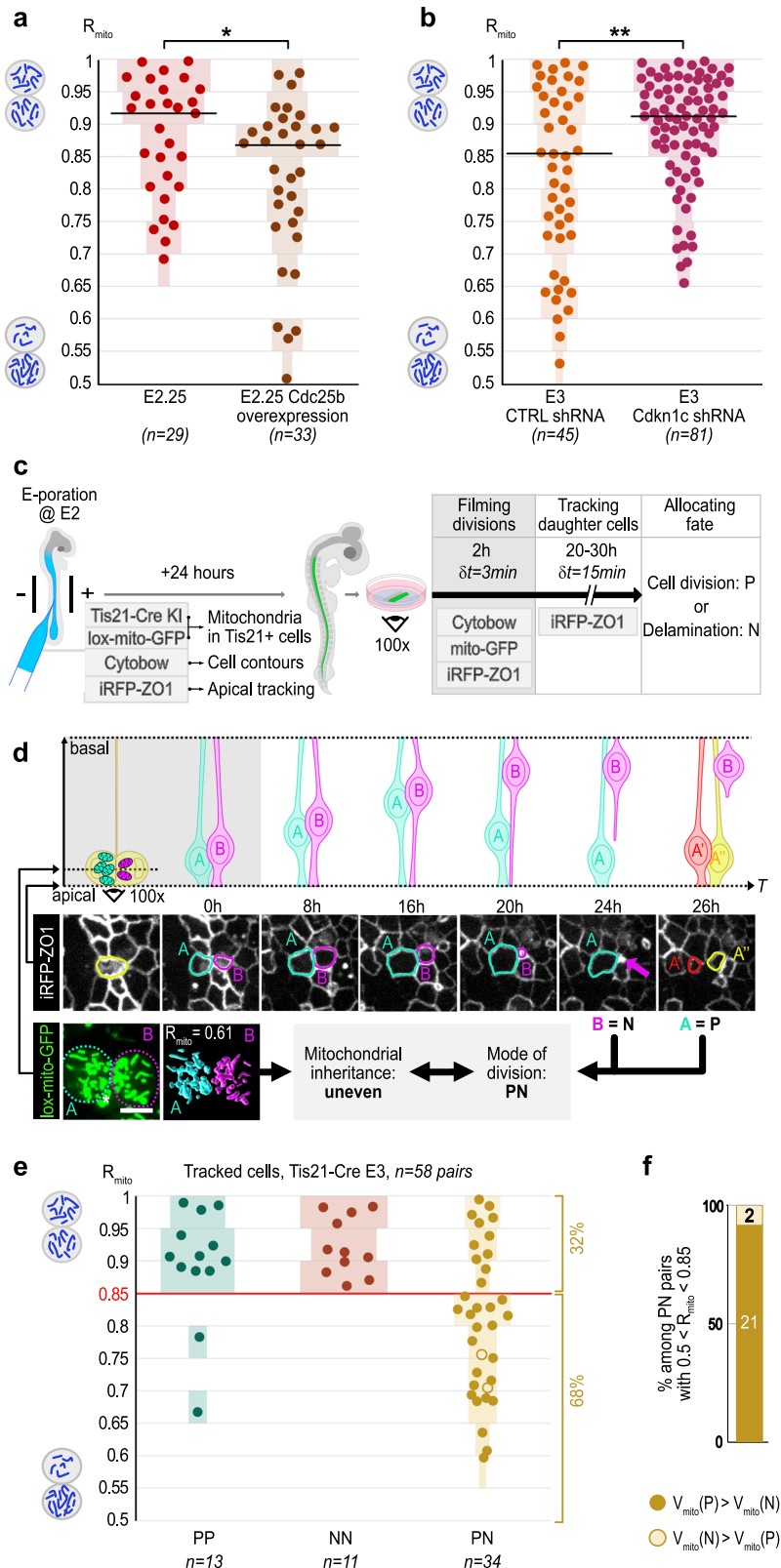

shows that an unequal mitochondrial inheritance is a strong predictor of an asymmetric fate decision. Individual tracking of pairs of cells further revealed a compelling near-exclusive relationship between mitochondrial pool size and fate: in 21 out of the 23 PN divisions showing unequal mitochondrial inheritance, the sister cell receiving fewer mitochondria differentiated into a neuron (Fig. 2e, f).

Our analysis of pairs of sister cells, and in particular the striking 'directionality' of unequal mitochondrial distribution that we observed during asymmetric neurogenic division, suggests that the unequal partitioning of mitochondria during mitosis initiates different mitochondria-regulated programs between progenitors and neurons during embryonic neurogenesis.

**Fig. 2 | Unequal mitochondrial inheritance by sister cells is a hallmark of asymmetric division. a** Scatter plots of $R_{mito}$ in sister pairs of control cells and upon overexpression of Cdc25b at E2.25. Two-tailed Mann-Whitney test. *: $p = 0.0199$. $n = 3$ embryos for control cells and $n = 4$ embryos for cells upon over-expression of Cdc25b. **b** Scatter plots of $R_{mito}$ in control (CTRL shRNA) cells and upon downregulation of *Cdkn1c* (*Cdkn1c* shRNA) at E3. Two-tailed Mann-Whitney test. **: $p = 0.003$. Light color bars in A and B represent the percentage of pairs in each 0.05 interval. $n = 4$ embryos for control cells and $n = 4$ embryos for *Cdkn1c* downregulation. **c** Experimental strategy combining live monitoring of mitochondrial inheritance and daughters' fate tracking. **d** Time-lapse series linking mitochondrial distribution and mode of division in Tis21 positive progenitors. Scheme (top) and en-face view (middle) of the time course from progenitor division to daughters' fate determination (division: progenitor, delamination: neuron). Bottom: unbalanced mitochondrial segregation in the mother cell ($R_{mito} = 0.61$)

matches with an asymmetric fate in sister cells. The asterisk marks mitochondria belonging to a neighboring cell and which are not included in the 3D reconstruction of the sister cell pair. Scale bar: 5 μm. **e** Scatter plots summarizing $R_{mito}$ and daughter fate (P = progenitor, N = neuron) from Tis21-positive progenitors. Red line: ($R_{mito} = 0.85$) proposed threshold separating equal from unequal inheritance. Open circles mark the only two cases of PN pairs with $R_{mito} < 0.85$, where the largest mitochondrial pool is inherited by the future neuron. Data from $n = 5$ embryos. **f** Inheritance of the smallest mitochondrial pool by the future neuron (full dots) or progenitor (open circles) in 23 PN pairs with unequal mitochondrial segregation. E = Embryonic day; T = time; δt = interval between time-lapse frames; h = hours; min = minutes; V = mitochondrial volume. Gray shading in panels c and d represents the period of high temporal resolution imaging (δt = 3 min). Source data are provided as a Source Data file.

## Chemogenetic manipulation of mitochondrial segregation during mitosis

We then investigated whether the relationship between unequal mitochondrial inheritance and differential fate acquisition is causal. To do so, we set up a chemogenetic approach aiming to directly manipulate mitochondria in order to induce their unequal inheritance in PP progenitors that normally distribute them equally, and subsequently assess the impact on the identity of their progeny.

In cultured cell lines, mitochondrial partitioning during mitosis is normally symmetric[20–22]. Strikingly, artificially tethering mitochondria to a kinesin in order to force their attachment to the microtubule network results in their asymmetric distribution to daughter cells[20]. We adapted this strategy to force the attachment of mitochondria to the microtubule network in neural progenitors in vivo. For this, we used the CatchFire (chemically assisted tethering of chimera by fluorogenic-induced recognition) method[23], to induce the rapid association of mitochondria with a plus-end directed kinesin motor upon application and removal of a specific ligand (Fig. 3a). The reversibility of this system allows the temporal control of this association. Thus, we can restrict the period of forced attachment to the duration of progenitor mitosis and avoid long-lasting effects on mitochondrial motility in their progeny.

We electroporated vectors expressing fusions of the CatchFire dimer moieties, one targeted to the outer mitochondrial surface (MitoFireMate) and the other fused to a kinesin motor domain (Kif-FireTag), (Fig. 3a) together with fluorescent mitochondrial and membrane reporters, and assessed mitochondrial behavior following administration of the ligand in en-face live preparations. In mitotic cells, within 15 min of ligand exposure, mitochondria appeared less dispersed and located more basally than in controls from metaphase until cytokinesis (Supplementary Fig. 5a, Fig. 6, and Supplementary Movies 6, 7). Most importantly, we observed unequal mitochondrial segregation during mitosis, including extreme $R_{mito}$ values lower than 0.5 (Supplementary Fig. 6b). The mechanism leading to the unequal segregation itself is unclear. It may simply be the result of the clustering of mitochondria creating a congestion near the cleavage plane that would, in turn, perturb their equal partitioning at the time of cytokinesis. Alternatively, the artificial tethering of mitochondria to microtubules may reveal or create by itself an asymmetry in the microtubule network, resulting in their directional transport. A gross disruption of the microtubule network that would perturb the normal mechanism of equal mitotic mitochondrial partitioning at E2 is not supported by experiments showing no visible alterations of the network integrity in mitosis in the CatchFire condition (Supplementary Fig. 5b).

We then calibrated the ligand administration regime to obtain $R_{mito}$ values contained within the physiological range observed in asymmetrically dividing progenitors, and to restrict the effect of the ligand to the approximate duration of mitosis (see "Methods" for details). With this optimized protocol, we monitored mitotic events at

E2, when a majority of progenitors undergo PP divisions and mitochondrial inheritance is mostly equal (Fig. 1d). We observed a significant shift towards unequal inheritance in ligand-treated cells, whereas the great majority of cells dividing before and after ligand exposure displayed the characteristic equal mitochondrial segregation normally observed at that stage (Fig. 3b, Supplementary Fig. 6c and see also Fig. 1d). Of note, the cells that displayed a $R_{mito} > 0.85$ during ligand exposure did not show any visible modification of mitochondrial dispersion (Supplementary Fig. 6e, f). This suggests that these mito-Cherry positive cells did not receive one of the two components of the CatchFire system, as can happen when a set of multiple vectors (here a set of four vectors) is electroporated[24].

The range of experimentally-induced $R_{mito}$ values obtained at E2 with this molecular toolkit is comparable to what is normally observed at E3 (Fig. 1d). We conclude that the CatchFire approach allows for efficient manipulation of the level of unequal mitochondrial distribution in neural progenitors in vivo.

## Forcing unequal mitochondrial segregation drives asymmetric fate

We next combined CatchFire manipulation of mitochondrial distribution with daughter cell fate tracking. We first monitored mitochondrial distribution in progenitors undergoing mitosis during the 1 h period of CatchFire ligand exposure. We then used two complementary approaches to establish cell fates (Fig. 3c). In the first one, we tracked daughter cells for 24 to 30 h, allowing identity to be assigned from a new round of mitosis or an apical detachment, as performed previously (Fig. 2c, d). In the second approach, we imaged daughter cells for only 2.5 additional hours, then fixed the samples and used molecular characterization to assign an identity. We took advantage of the specific expression of the hyperphosphorylated form of the Retinoblastoma protein (pRb) in cycling progenitors to assess the fate of daughter cells. At the end of the 2.5 h tracking period (see our characterization of pRb immunoreactivity in the "Methods" section), pairs of sister cells can be identified as PP, PN, or NN based on pRb positivity (P) or not (N) (Fig. 3d, Supplementary Fig. 7 and Supplementary Movies 8–10).

In these experiments, the increase in the number of pairs showing unequal mitochondrial segregation in the CatchFire condition (Fig. 3e, Supplementary Fig. 8a) matched with a rise in the proportion of asymmetric PN divisions, with more than half of the pairs (53%, $n = 19/36$) showing one sister cell committed to differentiation, compared to only 20% ($n = 8/40$) in the control condition (Fig. 3F, Supplementary Fig. 8b). This premature shift toward a neuronal fate in one of the daughter cells was observed specifically in pairs of sister cells displaying unequal mitochondrial inheritance (Fig. 3g, Supplementary Fig. 8c).

The shift to asymmetric fates was even more compelling when focusing on the 15 pairs (squares in Fig. 3g) with CatchFire-induced modification of mitochondrial dispersion (measured by a dispersion

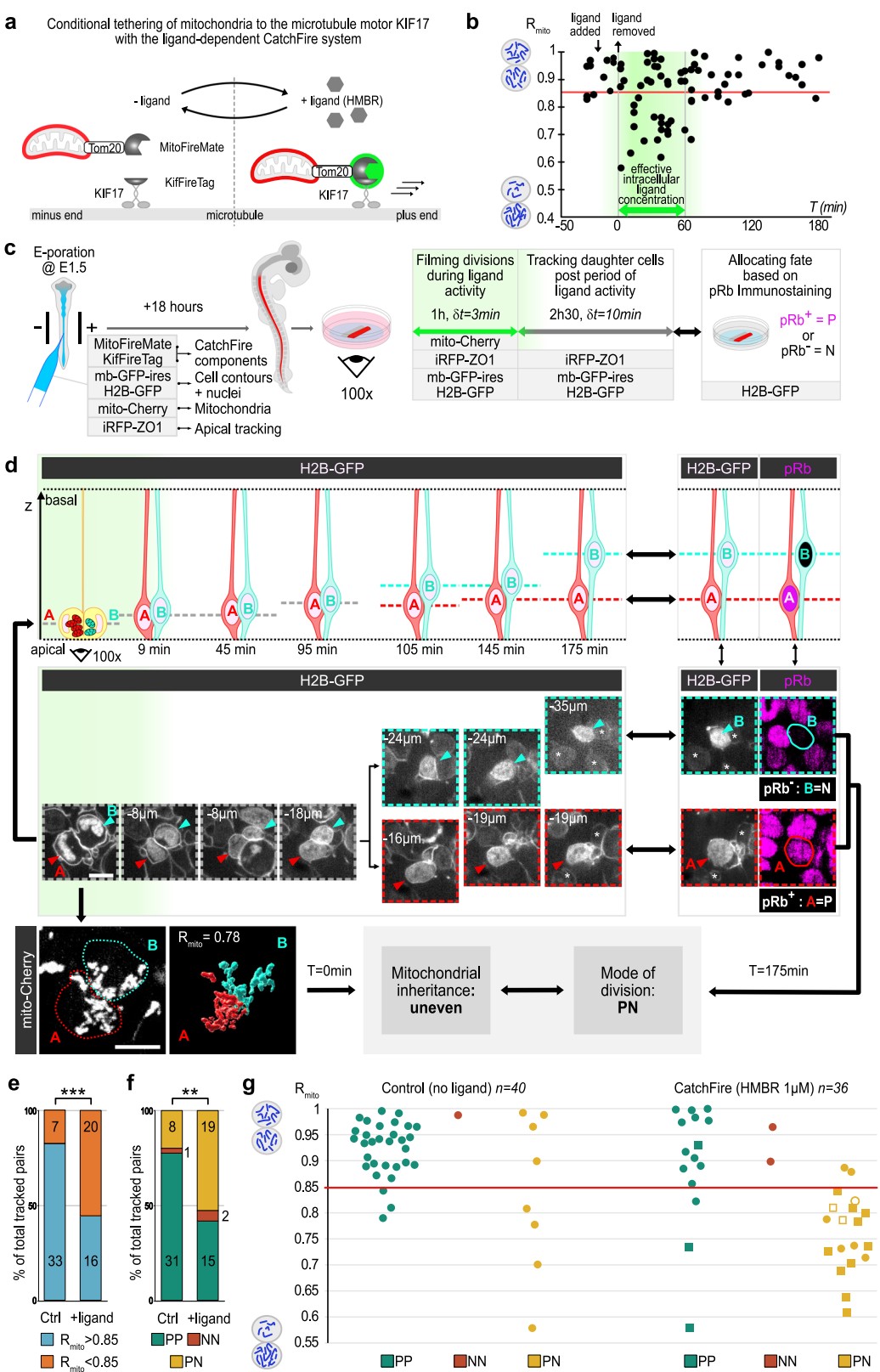

index (Di), see Supplementary Fig. 6 and "Methods") during mother cell mitosis: 14 of these pairs displayed unequal mitochondrial inheritance, and we observed a different sister fate (PN) in 12 of them (86%, $n = 12/14$; Fig. 3g). Additionally, the relationship between mitochondrial volume inheritance and fate determination was conserved: the smaller mitochondrial pool was inherited by the future neuron in 10 of these 12 PN pairs (Fig. 3g), consistent with our observations in asymmetrically dividing progenitors at E3 (Fig. 2e).

Overall, these results show that an experimentally induced mitochondrial imbalance during division is sufficient to induce a neurogenic fate in the daughter cell that inherits fewer mitochondria.

**Fig. 3 | Forced unequal mitochondrial inheritance induces asymmetric fate choices. a** Scheme of the CatchFire strategy to force mitochondrial attachment to the microtubular network during mitosis. **b** Scatter plots of $R_{mito}$ in CatchFire condition before, with (double arrow) and after washout of ligand. Green box: estimated period of ligand activity in cells. $n = 80$ pairs from 4 embryos. **c** Scheme of combined live monitoring of mitochondrial inheritance and daughter cell fate allocation by immunolabeling at E2.5. **d** Scheme (top) and en-face views (middle) of daughter cell's nucleus (H2B-GFP, white) in the depth of the neuroepithelium during the live tracking period (left) and correspondence with pRb immuno-fluorescence at its end (right, magenta). Images corresponding to z-levels of the two sister cells are color-coded (dotted frame) to illustrate the correspondence with the schematics of the time course. Bottom: $R_{mito} = 0.78$ in the mother cell matches with PN fate: Scale bar: 5 μm. Asterisks: neighboring cells used for registration between live and fixed images. **e, f** Representations of $R_{mito}$ and/or cell fate data from control and CatchFire pairs of cells obtained from the two tracking approaches. $n = 4$ embryos for control and $n = 4$ embryos for CatchFire pairs. Percentages of $R_{mito} < 0.85$ vs $> 0.85$ (**e**) and fate (**f**). Numbers in columns represent the number of pairs. Two-tailed Chi-2 test: (**e**) $p = 0.0005$; (**f**) $p = 0.006$. **g** Scatter plots summarizing $R_{mito}$ and daughter fate data. Squares in the CatchFire condition indicate a mitochondrial dispersion index (Di) < 0.6 (see Supplementary Fig. 6 and "Methods"). Note that, similar to our observations at E3 (Fig. 2e), the majority of PP pairs in the control condition at E2 (28/31) show a $R_{mito} > 0.85$. Open symbols in PN pairs: the smallest pool inherited by the progenitor. P = progenitor, N = neuron. E = Embryonic day; T = time; δt = interval between time-lapse frames; h = hours; min = minutes; P = progenitor; N = neuron. Green shading in panels (**b–d**) represents the deduced period of ligand effect. Source data are provided as a Source Data file.

## Discussion

Our experimental manipulation of mitochondrial distribution during mitosis, combined with fate tracking in vertebrate neural progenitors in vivo, represents the first direct causal demonstration that mitochondria act as asymmetric cell fate determinants. Previous work in several models of asymmetric division has correlated the description of an asymmetric inheritance of mitochondria with distinct morphologies or functions during mitosis to functional data showing their involvement in divergent fates in the progeny[3–8]. These studies have pointed to a role of differential qualitative characteristics of the inherited mitochondrial pools in cell fate acquisition. By contrast, our work identifies the importance of inheriting quantitatively unequal mitochondrial pools in the establishment of a different fate between sister cells. In particular, the CatchFire experiments show that an experimentally induced imbalance in the inheritance of mitochondrial volume drives neural differentiation during the progenitor amplification phase. While our results place the quantitative parameter at the heart of the asymmetric division mechanism, additional qualitative parameters within the inherited mitochondrial pools may also contribute to the final fate decision. In particular, this might be the case as more committed progenitors progressively acquire mitochondrial functional characteristics associated with differentiation[25,26]. These qualitative parameters could play a key role in the subset of PN pairs in our study where the inheritance was quantitatively balanced (Fig. 2e).

Our fate tracking data show that when mitochondrial segregation is unequal, the daughter cell inheriting a smaller mitochondrial pool consistently adopts a neuronal fate while its sister remains a progenitor. Inheriting fewer mitochondria may cause an energetic or metabolic deficiency that would, in turn, cell-autonomously trigger the post-mitotic remodeling of the mitochondrial network that regulates proneural versus proliferative programs (e.g.[9,10,27,28]). In this case, one would expect that future neurons should overall inherit a smaller absolute mitochondrial volume than cells that retain proliferative capacity. In line with this, one should also observe a progressive decrease in the mitochondrial volume between PP, PN and NN mother cells. However, analysis of the distribution of absolute mitochondrial volumes in fate-tracked daughter cells at E3 shows that the inherited volumes were homogenously distributed within a two-fold range and, more importantly, that this distribution was indistinguishable between P and N populations (Supplementary Fig. 9a). A similar overlap of the volume distribution in P versus N individual daughter cells was observed in the CatchFire experiments at E2 (Supplementary Fig. 9b). Consistent with these data, the absolute mitochondrial volume in mother cells (as deduced from the sum of volumes inherited by their daughters) showed a similar dispersion and was undistinguishable between the three modes of division (Supplementary Fig. 9c, d). This suggests that absolute mitochondrial amount is not sufficient to determine cell fate autonomously.

In contrast, the relative mitochondrial volume inherited by sister cells (illustrated by $R_{mito}$ values) is the parameter that correlates most strongly with, and best predicts fate outcomes. This opens the possibility that rather than instructing fate cell-autonomously through inherent quantitative and/or qualitative differences, unbalanced mitochondrial inheritance may act by modulating intercellular signaling between sisters. One attractive candidate is the Notch/Delta signaling pathway, which is involved in progenitor versus neuronal fate decisions during neurogenesis[29]. As mitochondrial activity has known links with the Notch/Delta signaling pathway[9,27], it is tempting to speculate that mitochondrial asymmetry might differentially regulate the ability of sister cells to send or receive Notch signals. This would initiate or amplify directional biases leading to binary fate choices.

Our in vivo data establish a causal link between a relative imbalance in the inheritance of mitochondria and differential fate acquisition between sister cells. Previous studies have proposed other determinants of identity during vertebrate neurogenesis, amongst which several regulators of the Notch signaling pathway[18,30–34]. As shown here for mitochondrial segregation, the unequal partitioning of each of these determinants shows a strong, but imperfect correlation with divergent fate between sister cells when characterized with a cellular resolution. Future studies will be needed to decipher whether they contribute synergistically to robust cell fate choices.

How is unequal mitochondrial segregation achieved? One attractive hypothesis is that they use an asymmetry of the central spindle for their routing towards the end of mitosis, as described for the unequal partitioning of SARA-endosomes in Drosophila and zebrafish neural progenitors[31,35]. More generally, this could represent a shared mechanism coordinating the asymmetric transport of different organelles involved in fate decisions[30–34]. Finally, deciphering how multiple determinants using different routes for their unequal distribution are coordinated represents a promising avenue for understanding how cellular processes integrate into developmental decisions[18,30–33].

## Methods

### Embryos

JA57 chicken fertilized eggs were provided by EARL Morizeau (8 rue du Moulin, 28190 Dangers, France). They were incubated at 38 °C in a Sanyo MIR-253 incubator for the appropriate time. The sex of the embryos was not determined. Under current European Union regulations, experiments on avian embryos between 2 and 4 days in ovo are not subject to restrictions.

### *In ovo* electroporation

*In ovo* electroporations in the chick neural tube were performed at embryonic day 1.5 (E1.5, HH stage 10) or E2 (E2, HH stage 14). The DNA solution, containing the vital dye FastGreen for visualization, was injected directly into the lumen of the neural tube via glass capillaries. Five 50 ms pulses of 20 V (at E1.5) or 25 V (at E2) separated by 100 ms intervals were applied using a square-wave electroporator (Nepa Gene, CUY21SC) and a pair of 5 mm gold-plated electrodes (BTX Genetrode

model 512) separated by a 4 mm interval. The egg was then sealed with parafilm and returned to the incubator.

## Plasmids

A list of plasmids used in this study is provided in Supplementary Table I. Plasmids and their full sequences are available upon request to X.M.

**Mitochondrial reporters.** The Mitobow construct (X503) has been described in[36]. pCX-lox-H2B-EBFP2-lox-mitoGFP (X1017, lox-mito-GFP in Figures) was created as follows: a pCX-lox-H2B-EBFP2-lox-mitoCerulean vector (X992) was first generated by deletion of a SacI-PmlI fragment from the Mitobow construct. A lox-H2B-EBFP2-polyA-lox-mito PCR fragment was then amplified from X992 with the following oligonucleotides (CX-fw 5'-GTCTCATCATTTTGGCAAAGAATTCTAGCT-3' and GFP-Eco-mito-rev 5'-CCTTGCTCACCATGGTGGCGAATTCGGAGGTGGCGACGG-3') and Gibson cloned into NheI/AgeI digested X-013 (pCX-GFP[37]).

pCX-mito-Cherry (X1202, mito-Cherry in Figures) was created as follows: a pCX-lox-mito-Cherry-lox-mito-Cerulean vector (X993) was first generated by deletion of a SmaI-PacI fragment from the Mitobow construct. A lox-mito-Cherry-polyA-lox-mito PCR fragment was then amplified from X993 with CX-fw and GFP-Eco-mito-rev oligonucleotides and Gibson cloned into NheI/AgeI digested X-013 (pCX-GFP[37]) to construct X-1118. pCX-mito-Cherry (X1202) was obtained by removing the BsrGI (stop-lox-mito-GFP) fragment from X-1118.

**CatchFire plasmids.** pK-Tom20-FireMate and pK-Kif17a-FireTag (unpublished vectors related to[23], generously provided by Franck Perez and Octave Joliot) were modified for optimized expression in the chick neural tube by replacing the CMV promoter with a CAGGS (CX) promoter. To generate pCX-Tom20-FireMate (MitoFireMate in Figures), the CMV promoter in pK-Tom20-FireMate was replaced by the Nde1/Nhe1 CAGGS promoter fragment from pCX-MCS2[37]. To generate pCX-Kif17a-FireTag (KifFireTag in Figures), the CMV promoter in pK-Kif17a-FireTag was replaced by an Nde1/Mfe1 fragment from pCX-H2B-miRFP670.

**Membrane contours.** pCX-mb-GFP-ires-H2B-GFP, pCX-mb-iRFP670 (cell membranes) or Cytobow (cytoplasm).

**Apical staining.** pCX-ZO1-miRFP670.

**Challenge of mode of divisions.** PMP-Cdc25b, PMP-lacZ were kind gifts from Fabienne Pituello's Lab. The shCdkn1c vector (X1101[15]) contains a chick Cdkn1c miRNA under the control of the chick U6 promoter, H2BGFP under the control of the CAGGS promoter.

**Somatic knock-in of the Cre recombinase at the *Tis21* locus.** Somatic knock-in of the Cre recombinase coding sequence at the C-terminus of the *Tis21* locus was achieved via CRISPR-Cas9-based Homology-Directed Recombination (HDR), through co-electroporation of a donor vector (Tis21-P2A-Cre, X907) and a CRISPR/Cas9 vector expressing the Cas9 recombinase and a gRNA targeting the C-terminus of the *Tis21* locus (pCX-SpCas9-cTis21-gRNA2, X854). The *Tis21* gRNA targets the cttggcagtgtaaggacaag sequence on the antisense strand and cuts 11 bases downstream of the *Tis21* stop codon. It was cloned in the BpiI-linearized X330 vector as described[38] using a duplex of oligonucleotides Tis21-gRNA2-fw (5'-CACCGCTTGGCAGTGTAAGGACAAG-3') and Tis21-gRNA2-rev (5'-AAACCTTGTCCTTACACTGCCAAGC-3'). The Tis21-P2A-Cre donor vector consists of a P2A-Cre cassette flanked with long left (1010 bp) and right (971 bp) arms of homology to the C-terminal region of chick *Tis21* (bGalGal1.mat.broiler.GRCg7b genome assembly). The P2A pseudo-cleavage sequence ensures that the Tis21 protein and Cre recombinase are produced as two independent proteins. In order to prevent targeting of the knock-in plasmid and re-targeting of the locus after insertion of the knock-in cassette by the gRNA, the gRNA target sequence was destroyed in the right arm of homology via the introduction of 7 base changes. Full details of the construction of this vector and of its validation as a bona fide *Tis21* targeting vector are described in[14].

Plasmid concentrations for monitoring the mitotic mitochondrial distribution at different stages of neurogenesis and functional challenges:

- for E2 experiments, pCX-mb-GFP-ires-H2B-GFP (0.5 µg/µl) and pCX-mito-Cherry (0.3 µg/µl) were electroporated at E1.5.
- for E3 experiments, either pCX-mito-CFP (0.5 µg/µl) or a mix of Mitobow (0.5 µg/µl) and pCX-Cre (0.1 µg/µl) were electroporated with pCX-mb-iRFP670 (0.3 µg/µl) at E2.
- for measurements in Tis21 positive progenitors at E3, Tis21-P2A-Cre donor and pCX-SpCas9-cTis21-gRNA2 vectors were each used at 0.8 µg/µl in the electroporation mix. The pCX-lox-H2B-EBFP2-lox-mito-GFP reporter plasmid (0.3 µg/µl) and pCX-mb-iRFP670 (0.5 µg/µl) were added to the electroporation mix.
- for experiments forcing premature neurogenic divisions at E2.25, pCX-mb-GFP-ires-H2B-GFP (0.5 µg/µl), pCX-mito-Cherry (0.3 µg/µl) and either PMP-Cdc25b (1 µg/µl) or the control vector PMP-lacZ (1 µg/µl) were electroporated at 1.5.
- for experiments forcing symmetric proliferative divisions at E3, pCX-mb-iRFP670 (0.5 µg/µl), pCX-mito-Cherry (0.3 µg/µl) and shCdkn1c (1 µg/µl) were electroporated at E2.

Plasmid concentrations for neurogenesis rate measurements: E1.5 embryos were electroporated with pCX-H2B-GFP (0.3 µg/µl) and either PMP-Cdc25b (1 µg/µl) or PMP-lacZ (1 µg/µl) for controls, and they were harvested 30 h after electroporation.

Plasmid concentration for fate tracking in Tis21+ progenitors: E2 embryos were electroporated with Cytobow (0.3 µg/µl), pCX-lox-H2B-EBFP2-lox-mito-GFP (0.3 µg/µl), pCX-ZO1-miRFP670 (0.3 µg/µl), Tis21-P2A-Cre (0.8 µg/µl) and pCX-SpCas9-cTis21-gRNA2 (0.8 µg/µl).

Plasmid concentration for CatchFire experiments: E1.5 embryos were electroporated with pCX-Tom20-FireMate (0.3 µg/µl), pCX-Kif17-FireTag (0.3 µg/µl), pCX-mb-GFP-ires-H2B-GFP (0.4 µg/µl), pCX-mito-Cherry (0.4 µg/µl) and pCX-ZO1-miRFP670 (0.4 µg/µl).

## Imaging

All the images in this study were obtained on an inverted microscope (Nikon Ti Eclipse) equipped with a spinning disk confocal head (Yokogawa CSUW1) with Borealis system (Andor) and a sCMOS Camera (Orca Flash4LT, Hamamatsu) and the following Nikon objectives: 10x (CFI Plan APO λ, 0.45), 40x (CFI Plan APO λ, 0.95) or 100x (CFI APO VC, NA 1.4, oil immersion). The setup is driven by MicroManager software and equipped with a heating enclosure (DigitalPixel, UK) set to 38 °C for live imaging experiments.

## En-face culture

En-face culture of the embryonic neuroepithelium was performed at E2.25 or E3. After extraction from the egg and removal of extra-embryonic membranes in 1xPBS (thereafter PBS), embryos were transferred to 38 °C F12 medium and pinned down with dissection needles at the level of the hindbrain in a 35 mm Sylgard dissection dish. A dissection needle was used to slit the roof plate and separate the neural tube from the somites from the hindbrain to the caudal end on both sides of the embryo. The neural tube and notochord were then transferred in a drop of F12 medium to a glass-bottom culture dish (MatTek, P35G-0-14-C), and medium was replaced with 1 ml of 1% low-melting-point (LMP) agarose/F12 medium (maintained at 38 °C). Excess medium was removed so that the neural tube would flatten with its apical surface facing the bottom of the dish, in an inverted open book conformation. After 30 s of polymerization on ice, an extra layer

of agarose medium (200 μl) was added to cover the whole tissue and left to harden. 2 ml of 38 °C culture medium was added (F12/1 mM Sodium pyruvate), and the culture dish was transferred to the 38 °C chamber of a spinning disk confocal microscope.

### Live monitoring of mitochondria during cell divisions

Several full-frame fields of view (136*136 μm) were selected on the flat-mounted neural tube, and ~25 μm deep z-stacks were acquired with a 0.3 μm z-step size at 3 min intervals during 1 to 2 h. Images were saved as separate images and imported into Fiji. Pairs of daughter cells at the first time point after division, presenting both membrane or cyto-plasmic reporter signal and mitochondrial signal, were identified and selected to be cropped on their entire height. Selected crops were saved as image stack files for 3D reconstruction (see below).

### Daughter fate tracking experiments

For fate tracking experiments, an initial acquisition of images was performed using the parameters described above for a duration of 1 to 2 h. During this period, we monitored mitochondrial inheritance in dividing mother cells and started to track their daughters. We then switched to long-term tracking using different acquisition parameters as detailed below.

Of note, mitochondrial ratio at cytokinesis and corresponding daughter fates were determined independently in a blind manner within the whole set of pairs. Mode of division and mitochondrial inheritance were matched retrospectively.

For long-term live tracking, the acquisition parameters were modified as follows: ~25 μm deep z-stacks centered around the apical surface were acquired with a 1 μm z-step at 15 min intervals for 24 to 30 h. To reduce potential phototoxicity, the mitochondrial channel was omitted during this sequence as it is not required for tracking. Using the apical surface marker (ZO1-miRFP670), we were then able to track daughter cells with long term imaging and assign their identity from their behavior as follows: a delamination process characterized by a progressive constriction (over at least three consecutive time points) of the apical surface followed by apical detachment for a future neuron[17–19], versus apical mitosis for a progenitor. Of note, some daughter cells detach abruptly (within one or two time points) from the apical surface a few hours after mitosis without displaying the characteristic progressive shrinkage of the apical surface described above. These cells are probably dying and have not been considered in the analysis. The purpose of keeping a large z-depth centered on the apical surface was to accommodate for potential z-drifting of the embryo relative to the coverslip during the long acquisition.

For tracking experiments combining short-term live tracking and immunostaining, we used the following protocol: we increased the depth of imaging to 50 μm, and acquired 1 μm-spaced z-stacks every 10 or 15 min for 2 h to 2h30. We then switched to the 10x objective and acquired an extended field of view to acquire spatial landmarks in order to facilitate the repositioning of the culture dish with the same orientation after immunostaining. The culture dish was then removed from the microscope stage and immediately processed for fixation and immunostaining. The whole staining procedure was performed at 4 °C, and we only subjected the samples to very gentle agitation to ensure that the neural tubes remained embedded in the 1% LMP agarose layer. The culture medium was removed, and the samples were quickly washed once in PBS before fixation for 90 min in ice-cold 4% paraformaldehyde (PFA) in PBS, followed by 3 washes of 15 min in PBS. The neural tube was incubated for 48 h with the anti-pRb and anti-GFP primary antibodies in the blocking solution (PBS/0,3% Triton/10%Fetal Calf Serum (FCS)). The samples were then washed 3 times for 15 min in PBS and incubated for 48 h in the dark with the secondary A488 anti-chicken (1/500) and Cy5 anti-rabbit (1/500) antibodies and DAPI (1/1000) in PBS/0,3% Triton. The samples were then washed 3 times for 15 min in PBS. 1 ml of PBS was added to the culture dish, which was

subsequently transferred to the microscope. We first used the 10x objective to reposition the dish in the same orientation and recover each original field thanks to the 10x images acquired at the end of the 2h30 movie. We then switched to the 100x objective and used the last images of the 100x live sequence to precisely recover the corresponding fields and acquire matching 4-channel z-stacks in the immune-stained neural tube (405 nm/DAPI; 490 nm/mb-GFP and H2B-GFP; 561 nm/mito-Cherry; 641 nm/Cy5-pRb).

We then used the H2B-GFP signal in the live sequence to track daughter cells as their nucleus starts to move basally after mitosis. Briefly, for each field of view, we imported the stack files of the 1 h and 2 h30 live sequences in Fiji. We identified divisions of electroporated progenitors occurring during the high-resolution 1 h movie, and we followed the nucleus of the daughters during the 2 h30 tracking movie. For each pair in which we were still able to visualize the nucleus in each sister cell at the last time point, we proceeded to analyse the corresponding 100x z-stack from the fixed sample and recovered these sister cells on the basis of their H2B-GFP signal. To distinguish between cycling progenitors and postmitotic neurons, we used pRb immunoreactivity, as we have established that 2.5 h after mitosis, it is a discriminating factor between these two cell populations (pRb (+) = progenitor; pRb (-) = neuron).

This 2.5-h time window was established beforehand in E2 embryos as follows: we determined the time point after mitosis at which all progenitors in a synchronized cohort of cells undergoing mitosis have crossed the restriction point/late G1 stage, as determined by pRb immunoreactivity. We used the cell-permeant dye FlashTag (FT) to label a cohort of pairs of sister cells that perform their division synchronously at E2, and counted the proportion of pRb-positive cells in the cohort in transverse sections from embryos harvested at different time points after FT injection. This proportion should reach a plateau when all the progenitors in the cohort have passed the restriction point and have become positive for pRb. At E2, this plateau was reached between 1.5 h and 2 h after injection, and the distribution of pRb immunoreactivity in FT-positive cells remained stable at later time points (2.5 h and 3 h). This indicates that from 2 h after mitosis injection, pRb-positivity is a reliable marker of progenitor status, and that a pair of cells containing 0, 1, or 2 pRb-positive cells corresponds to an NN, PN and PP pair, respectively. For these experiments, a 1 mM stock solution of FlashTag was prepared by adding 20 μl of DMSO to a CellTrace Far red dye stock vial (Life Technologies, #C34564)[19]. A working solution of 100 μM was subsequently prepared by diluting 1 μl of stock solution in 9 μl of 38 °C pre-heated PBS, and injected directly into E2 chick neural tubes. The eggs were resealed with parafilm, and embryos were incubated at 38 °C for the appropriate time until dissection.

### 3D reconstruction and measurement of cellular and mitochondrial volumes

Images of pairs of daughter cells at the first time point after division were imported as stacks in IMARIS software (Bitplane) to perform 3D reconstructions of cell volume and mitochondrial volume. Each daughter cell was individually segmented and reconstructed. Cells were delineated by drawing their contour on each z plane through their entire height using either the membrane reporter signal or a cytoplasmic reporter signal. With the contouring of the daughter cells, the cellular volume of each daughter cell was reconstructed using the surface creation tool of IMARIS. The mitochondrial signal contained in the cellular volume of each daughter cell was then isolated by masking the mitochondrial signal located outside of this volume. Each daughter's mitochondrial volume was then reconstructed with the surface creation tool, using the same threshold for both cells. Data of both mitochondrial and cellular volumes for each daughter cells were then used to perform ratio measurements to assess the mitochondrial distribution between daughter cells and to compare their cellular volume.

Proposed $R_{mito}$ threshold to distinguish equal versus unequal mitochondrial segregation: based on $R_{mito}$ distribution in symmetric proliferative and neurogenic terminal dividing progenitor observed in fate tracking experiments (Fig. 2), we calculated the first decile, which correspond to 0.85, as a threshold value to discriminate between equal versus unequal mitochondrial segregation.

Proposed $R_{mito}$ threshold to distinguish symmetric versus asymmetric fate: we also estimated the optimal $R_{mito}$ threshold value that would best predict symmetric versus asymmetric fates in the tracked Tis21-Cre data set at E3. For this, we fit a Classification Tree[39] with a max depth of 1, with the fate (symmetric or not) as a binary outcome and the proportion of inherited mitochondria ($R_{mito}$) as the only covariate. This algorithm looks for the best value that splits the dataset into two subsets for which the impurity (chosen as the Gini impurity here) is minimized. To compute a 95% confidence interval for this threshold, we bootstrapped the initial dataset ($N = 1000$ iterations). The algorithm incorporates a weighing parameter for the proportion of PP, PN and NN divisions in sampling. Importantly, using the observed distribution of the three modes of divisions in our dataset (22%, 59% and 19%, see Fig. 2), or their expected distribution in the tissue at that stage (respectively 50%, 34% and 16%[14],) returned the same value of 0,85 (respectively 0.853, 95% CI 0.75-0.88 and 0.853, 95% CI 0.84-0.89).

## Forced attachment of mitochondria to kinesin using CatchFire

We developed a time-controlled strategy to reversibly force mitochondrial attachment to the microtubular network (Fig. 3a). The rationale was to restrict the perturbation to a time window around progenitor division in order to avoid long-lasting effects on mitochondrial motility that may perturb their activity and be deleterious for daughter cell survival. The recently developed CatchFire (chemically assisted tethering of chimera by fluorogenic-induced recognition) method is based on the reversible ligand-dependent dimerization of small genetically encoded protein partners. The smaller moiety of the dimer (FireTag) was fused to the N-terminus of the motor domain of the plus-end kinesin-like KIF17[23] (thereafter called KifFireTag). The larger moiety (FireMate) was anchored to mitochondria via the targeting sequence of the outer mitochondrial membrane protein Tom 20 (MitoFireMate). The two partners can therefore enable the rapid and reversible association of mitochondria with the microtubule motor upon ligand administration and removal, respectively (Fig. 3a).

Embryos were electroporated at E1.5 with the appropriate plasmid combination (see above) and processed 18 h later for en-face live-imaging according to the procedure described above. To induce the pairing of KifFireTag and MitoFireMate, a 20 µM solution of ligand HMBR was prepared by diluting the original 20 mM stocks in F12 medium, and an appropriate volume was added to the culture dish to reach the desired final concentration of ligand. We first established a protocol of transient exposure of the sample to ligand-containing medium, that considers the diffusion kinetics of the ligand within the complex neuroepithelial tissue. Based on our previous experience with the HMBR ligand and pFAST, the fluorogenic monomeric parent of the CatchFire system, we expected that the diffusion of the ligand within the tissue would take between 15 and 30 min, and that it would display a slow disappearance kinetics after being washed out from the culture dish[40]. We therefore anticipated that the induction and cessation of pairing of the CatchFire partners would follow similar kinetics. We found that continuous exposure (1 h) and/or high ligand concentration (10 µM) led to aberrant mitochondrial distribution (Supplementary Fig. 6b) and mitosis failure in several dividing cells. After several tests, we determined an optimal HMBR concentration (1 µM) and duration of exposure (15 min) after which ligand removal from the culture medium initiates a period of progressive depletion of the ligand from neuroepithelial cells. With this protocol, the intracellular ligand concentration appears to remain in the range of efficient CatchFire pairing for approximately 1 h before the effect disappears (as judged from

measurements of mitochondrial inheritance and dispersion index—see Fig. 3b, Supplementary Fig. 6e).

In the final protocol used for fate tracking experiments, we started by selecting several fields of interest for imaging in the absence of the ligand. We then added ligand to the dish to reach a 1 µM concentration in the medium. After 15 min the ligand was removed from the culture dish by three gentle washes with 2 mL of HMBR-free F12 medium. We immediately started imaging progenitor divisions in the selected fields for 1 h using the acquisition parameters described above in the "Live monitoring of mitochondria during cell divisions" section. After the 1-h movie, we switched to the relevant acquisition parameters described above in the "Daughter fate tracking experiments" section.

For the live visualization of the microtubule network (Supplementary Fig. 5, Supplementary Fig. 6a), dissected neural tubes were incubated in a SiR-tubulin solution (SC006: Cytoskeleton Kit, tebu-bio; 1 µM final concentration in liquid F12 culture medium) during 1 h at 38 °C, they were then mounted for en-face culture in LMP agarose medium and incubated in SiR-tubulin at 100 nM final concentration in liquid F12 culture medium.

## Dispersion index measurement

The Dispersion index (Di) was determined using ImageJ software. Di quantifies the effect of CatchFire pairing on mitochondrial dispersion in dividing progenitors by measuring the ratio of a z-projection of the whole mitochondrial volume, imaged from the apical surface, to the surface of the maximum cell contour at the time cytokinesis (Supplementary Fig. 6d). The maximum cell contour of the pair of sister cells was manually outlined from a 3 µm z projection of the equator region where the daughter cell pair's surface area is maximum. Then, a z-projection of the total mitochondrial fluorescence from the pair of cells was performed, and contours of this fluorescence were manually drawn. For each pair of daughter cells, the ratio of the area occupied by mitochondria to the maximal projected area of the daughter cell pairs was calculated (Supplementary Fig. 6d). All control progenitors in the absence of ligand display ratios above 0.6 (Supplementary Fig. 6e). Accordingly, cells with a Di value below 0.6 were considered as having a "CatchFire effect" on mitochondrial distribution.

## Immunostaining on vibratome sections

For vibratome sections, chick embryos were electroporated at E1.5 and harvested 30 h later in ice-cold PBS. Extraembryonic membranes were quickly removed, and embryos were fixed for 1 h in ice-cold 4%PFA/PBS and rinsed 3 times for 5 min in PBS at room temperature (RT). The trunk region was then dissected and embedded in 4% agarose (4 g of agarose in 100 ml of water, boiled in a microwave and cooled at 50 °C). 100 µm vibratome sections were obtained using a Microm HM 650 V Microtome (ThermoScientific) and collected in 6-well plates in ice-cold PBS.

Vibratome sections were incubated for one day with the primary antibodies diluted in PBS/0.1%Triton (PBST) at 4 °C with gentle agitation. Sections were then washed 3 times for 5 min in PBST at room temperature (RT), incubated overnight in the dark at 4 °C with the appropriate secondary antibodies and DAPI diluted in PBST, washed again 3 times for 5 min at RT with PBST and mounted with Vectashield (Vector Laboratories H-1000-10). Primary antibodies used are: rabbit anti-pRb (Ser807/811–1:1000) from Cell Signaling; mouse anti-HuC/D (clone 16A11–1:50) from ThermoFisher Scientific; Chick anti-GFP (1:800) from Aves Labs (see Supplementary Table I for additional details). Alexa Fluor 488-coupled anti-chicken (1:500) and Cy5-coupled anti-mouse (1:500) and Cy3 anti-rabbit (1:500) secondary antibodies were obtained from Jackson Laboratories.

We explored the effect of overexpressing Cdc25b in progenitors on the production of neurons at the tissue level. In order to target and investigate specifically the neurogenic transition, we concentrated our analyses on the dorsal half of the neural tube where this transition has

barely started at the time of analysis (E2.75, HH17) whereas most ventral progenitors are already neurogenic. Neuron and progenitor populations were evaluated at 30 h after electroporation (hae) at E1.5 via immunofluorescence (GFP-pRb positive cells were considered as progenitor, GFP-HuC/D positive cells as neurons (Supplementary Fig. 3). Cell counts were performed on several distinct and non-overlapping z planes per vibratome section.

## Statistical analyses

The number of embryos and analysed cells or pairs of cells is indicated in the Legends to the Figures. All data processing and statistical analyses were performed using Excel and GraphPad Prism 6 software, and the tests are indicated in the Legends to the Figures. Statistical significance of $p$-values is represented on graphs as: ns, $p > 0.05$; *, $p < 0.05$; **, $p < 0.01$; ***, $p < 0.001$. Exact $p$-values for *, **, *** are provided in the Legends to Figures.

## Reporting summary

Further information on research design is available in the Nature Portfolio Reporting Summary linked to this article.

## Data availability

Source data are provided with this paper. Plasmids used in this study and their full sequences are available upon request to X.M. Source data are provided with this paper.

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

## Acknowledgments

We thank Samuel Tozer for insightful discussions on the project, and our colleagues Jean-François Brunet, Sonia Garel, Alice Meunier, Jonathan Weitzman, Marco Pontoglio, Arnaud Echard and Samuel Tozer for their comments on the manuscript. We thank Felix Cheysson for the bootstrapping method and calculation. We thank Fabienne Pituello for the kind gift of Cdc25b plasmids, and Octave Joliot and Franck Perez for generously sharing unpublished components of the CatchFire system. We thank the IBENS imaging platform for access to the IMARIS analysis station. This study was funded by Agence Nationale de la Recherche grant ANR-18-CE16-0021-01 SYMASYM (XM); Fondation pour la Recherche Médicale grant FRM EQU202003010547 (XM); Labex MEMOLIFE (XM); AFM-Téléthon Trampoline grant 28934 (EF and XM); Agence Nationale de la Recherche grant ANR-10-LABX-54 MEMO LIFE as part of IBENS.

## Author contributions

B.B., R.G., R.L., and E.F. performed experiments. B.B., R.L., A.G., X.M., and E.F. provided resources and methodology. E.F., B.B., R.L., and X.M. participated in designing experiments, data analysis, and data interpretation. E.F. and X.M. wrote the manuscript. All authors (B.B., R.L., R.G., A.G., X.M., and E.F.) provided input and revisions to successive drafts of the entire manuscript. E.F. conceived the project. E.F. and X.M. managed the overall project, and X.M. obtained funding.

## Competing interests

A.G. is listed as inventor on a patent application related to the CatchFire system and filed by Sorbonne Université, the École Normale Supérieure – PSL University, the CNRS and the Institut Curie. A.G. is also co-founder of The Twinkle Factory, a start-up that will commercially distribute the match molecules for research use. The remaining authors declare no competing interests.
