## [Peer Review file · Nature Communications]

Unequal mitochondrial segregation promotes asymmetric fates during neurogenesis

Corresponding Author: Dr Evelyne Fischer

Version 1:

Reviewer comments:

Reviewer #1

(Remarks to the Author)

This manuscript describes a novel phenomenon that influences cell fate decisions during neural progenitor cell divisions. The authors report that asymmetric inheritance of mitochondria during divisions results in asymmetric cell fates, with the cell receiving the smaller relative volume of mitochondria eventually acquiring neuronal fate. This manuscript is an impressive piece of work, relying on elegant and challenging in vivo live imaging experiments that clearly demonstrate the main conclusion of the paper. The observations made here are beautiful and clearly of high impact, with implications beyond neural development. The figures and illustrations are exceptionally well put together and informative. Notably, this is the first in vivo demonstration that unequal mitochondrial volume partitioning affects cell fate. These findings are noteworthy, but would be further strengthened through a more mechanistic examination of the process; however, we appreciate the challenges associated with investigating this. These findings will be of broad interest, including to those working on general cell biology, developmental biology, and neuronal development. We do, however, have the following comments and recommendations for the authors:

Introduction

- In the final paragraph of the introduction, the authors ask if “asymmetric mitochondrial behaviour within the dividing mother cell triggers the mitochondrial changes that control fate decisions in the progeny”; however, they only address whether asymmetric mitochondrial inheritance affects cell fate decisions and not how this affects changes in mitochondrial function.

- Fission and fusion are discussed in the introduction (line 35). How does this relate to the findings?

Results 1

- The authors refer to Mito-GFP being used as a marker for Tis21-Cre expressing progenitors, but there is no example of this shown in Fig. 1 or Extended data Fig. 2.

- The authors should define Rcell clearly in the text.

- Extended data Fig. 2, currently the x scale is more expanded than the y. Matching scales would be more informative.

Results 2

- Fig. 2B. It is unclear what is meant by the statement “Control cell values at E3 are the same as in Fig. 1D” as the data in these figures appear to be from distinct experiments.

- Fig. 2B. There is a double asterisk on the graph but only a single in the legend. Is the p value accurate?

- Fig. 2D. In the lox mito-GFP image, the border of cell A drawn on the image excludes mitochondria included in the reconstruction on the right. Which is accurate?

- Fig. 2E. These data are based on a good sample size of cells undergoing PN divisions; however, the number of PP and NN cell numbers are much lower. Given these differences in cell numbers, is it valid to compare differences in mean Rmito

between these populations? The authors may want to consider increasing their sample sizes for this experiment. The authors should also define open and filled dots in 2E, although later defined in 2F.

Results 3

- Extended data Fig. 3. The authors may want to consider revisiting this data and providing cell numbers. In the representative images of neural tube sections provided, the difference in neuron numbers is not clear. In fact, there appear to be more HuC/D expressing cells on the electroporated sides of the control embryo compared to the CDC25B expressing cells. It would also be helpful to outline the region of transfected neurons to aid comparison of electroporated cells only.

Results 5

- The authors should expand upon the significance of CatchFire mediated changes in mitochondrial dispersion and basal localisation and the rationale behind how this may lead to asymmetric inheritance of mitochondria.
- Is it possible that CatchFire could have caused ACD because of disruption to microtubules rather than mitochondria? This possibility should at least be discussed.

Discussion

- Discussion should be expanded with more critical interpretation of their findings.
- Line 218. They argue against qualitative differences in mitochondria affecting cell fate, suggesting that volume is important, but both could be true.

Reviewer #2

(Remarks to the Author)

Reviewer #3

(Remarks to the Author)

Bunel et al

This is an excellent and very well written manuscript that will have a major impact on our understanding of the role of mitochondria in cell fate decisions in the context of asymmetric cell division. The authors used state of the art technology, the experiments were done in a very rigorous way, and the data is convincing. Overall, I very much enjoyed reading it. This is great work, and the authors can be congratulated.

Below are a few comments and questions.

1. In Fig S2, the authors present data that Rmito and Rcell do not correlate. However, it is unclear why they did not consider cell volume in all of their data by determining mitochondrial volume per cell volume (mito/cell) for each daughter cell and then taking the ratio of that. I can see that both is of value, but it would be interesting to see how the data looks then, and it might hint at an interaction between mitochondria and the nucleus/genome or the cytoplasm/other organelles.

2. Along those lines, have the authors looked at the inheritance of other organelles in their system?

3. In Fig 1C, it appears that mitochondria in the 'even' example are distributed throughout the daughter cells whereas in the 'uneven' example, they are more organized, clustered around what was the division plane. In Fig S1 it seems the opposite, the uneven example seems more distributed throughout the cell and the even example more clustered. (In the CatchFire experiment and Fig S6, similar phenomena are observed and referred to as 'distribution'.) Can the authors elaborate on this? Could it be slightly different timepoints after cytokinesis? Or do the authors think there is control of distribution prior to cell division? As it stands, this is confusing.

4. Cell fate tracking and CatchFire experiment. In the abstract the authors state "We set up a chemogenetic approach to experimentally displace mitochondria specifically during mitosis to force their unequal inheritance in vivo and we found that this was sufficient to drive premature neuronal differentiation." However, they are more careful in the title, which is "Unequal mitochondrial segregation promotes asymmetric fates during neurogenesis". Data shown in Fig 2D, Fig. 3G and Fig S8C, suggest that low R mito (below 0.85) is neither required nor sufficient for asymmetric fates (P/N) (for example R mito of 0.65 in PP divisions and 32% of P/N divisions have R mito of 1-0.85). I agree with the authors that unequal mitochondrial segregation promotes P/N divisions but I think they need to be a bit more careful when using the term 'sufficient'. Based on their data, other cell fate determinants are likely to play a role and contribute, and this should be stated and discussed.

5. Figure 3 Parts C and D. I found this not easy to follow. Maybe those schematics could be simplified or the labelling changed.

6. The authors state that their data indicates that mitochondrial volume (quantity) rather than a qualitative aspect is important for asymmetric fate. However, they do not provide any evidence that there is no qualitative difference between mitochondria in P/N sisters. Have the authors analysed for example mitochondrial morphology?

7. Along these lines, the CatchFire system could potentially bias towards the tethering of small organelles. Have the authors seen any evidence of that? And could CatchFire impact the cell cycle by activating an organelle inheritance check point? Have the authors observed cell cycle delays?

Version 2:

Reviewer comments:

Reviewer #1

(Remarks to the Author)

We are happy that the authors have addressed all our comments and strongly recommend this manuscript for publication, with congratulations to the authors.

Reviewer #2

(Remarks to the Author)

Reviewer #3

(Remarks to the Author)

The authors have done a great job addressing my comments and concerns.

REVIEWER COMMENTS

Reviewer #1 (Remarks to the Author):

This manuscript describes a novel phenomenon that influences cell fate decisions during neural progenitor cell divisions. The authors report that asymmetric inheritance of mitochondria during divisions results in asymmetric cell fates, with the cell receiving the smaller relative volume of mitochondria eventually acquiring neuronal fate. This manuscript is an impressive piece of work, relying on elegant and challenging in vivo live imaging experiments that clearly demonstrate the main conclusion of the paper. The observations made here are beautiful and clearly of high impact, with implications beyond neural development. The figures and illustrations are exceptionally well put together and informative. Notably, this is the first in vivo demonstration that unequal mitochondrial volume partitioning affects cell fate. These findings are noteworthy, but would be further strengthened through a more mechanistic examination of the process; however, we appreciate the challenges associated with investigating this. These findings will be of broad interest, including to those working on general cell biology, developmental biology, and neuronal development. We do, however, have the following comments and recommendations for the authors:

Introduction

•In the final paragraph of the introduction, the authors ask if “asymmetric mitochondrial behaviour within the dividing mother cell triggers the mitochondrial changes that control fate decisions in the progeny”; however, they only address whether asymmetric mitochondrial inheritance affects cell fate decisions and not how this affects changes in mitochondrial function.

Based on the Reviewer’s comment, we have modified the sentence in the Introduction section as follows:

p3 line 23: “asymmetric mitochondrial behavior within the dividing mother cell triggers differential fate decisions in the progeny”

•Fission and fusion are discussed in the introduction (line 35). How does this relate to the findings?

In the introduction section, we stated that “Notably, fate decisions in the progeny of cortical progenitors depend on modifications in the balance between mitochondrial fusion and fission taking place within a brief time window following cell division¹⁰”. This referred to a study showing that experimental changes of this dynamics within a few hours after mitosis can influence cell fate. Our findings complement this study by focusing on events linked to mitochondria that occur during mitosis rather than in the daughter cells after mitosis.

The quantitative imbalance that we observed during mitosis could be related to the post-mitotic changes in the network in different ways. An easy “direct” hypothesis would have been that the absolute mitochondrial volume is “measured” by the daughter cell, and that it would in turn adapt by modifying its fusion/fission balance, leading to mitochondrial network remodeling in an autonomous fashion. However, our results highlight that the “relative” rather than the “absolute” mitochondrial volume is the key factor. This leads to the more likely and “indirect” hypothesis that the unequal inheritance initiates an intercellular dialog that may in turn trigger the post-mitotic remodeling of the mitochondrial network involved in the final fate decision.

In the course of this revision, we realized that in the sentence above, the use of the term "involves" rather than "depends on" describes more accurately the results of the study by Iwata et al., and propose to reword it as follows: "Notably, fate decisions in the progeny of cortical progenitors involve modifications in the balance between mitochondrial fusion and fission taking place within a brief time window following cell division¹⁰"

Results 1

•The authors refer to Mito-GFP being used as a marker for Tis21-Cre expressing progenitors, but there is no example of this shown in Fig. 1 or Extended data Fig. 2.

We have added an image of a Tis-CRE → mito-GFP mitotic cell in Fig. 1d, to complement the example that was already shown in Fig. 2d. The following sentence has been added to the legend: p30, line 748: "The inset shows an example of Tis-Cre driven mito-GFP in a mitotic progenitor in an en-face confocal view. Scale bar 5µm."

•The authors should define Rcell clearly in the text.

We have now defined Rcell more explicitly in the main text: p5 line 70

"To explore whether unequal mitochondrial volume inheritance is a consequence of unequal cell volume partitioning, we next monitored cellular volumes obtained by 3D reconstruction of cell contours in the daughter cells of Tis21-Cre progenitors after cytokinesis. We calculated a Rcell metric representing the ratio of the volume of the cell containing the smallest mitochondrial pool to the volume of its sister cell containing the largest mitochondrial pool (Supplementary Fig. 2). In contrast to the high frequency of uneven mitochondrial inheritance, the cellular volume was nearly identical between sister cells, as illustrated by Rcell values consistently close to 1 (Supplementary Fig. 2). The absence of correlation between Rmito and Rcell (Supplementary Fig. 2) indicates that unequal mitochondrial volume inheritance constitutes an intrinsic factor and is not the consequence of a differential cell volume."

In addition, we added the following comment in the legend to Supplementary Fig. 2 (top panel):

"Note that the directionality of the Rcell ratio is dictated by Rmito, and that as a consequence, Rcell values can be higher than 1"

•Extended data Fig. 2, currently the x scale is more expanded than the y. Matching scales would be more informative.

We have reformatted the Supplementary Fig. 2 to match the x and y scales.

•Fig. 2B. It is unclear what is meant by the statement "Control cell values at E3 are the same as in Fig. 1D" as the data in these figures appear to be from distinct experiments.

We thank the reviewer for spotting this sentence, which was added here by mistake. The control values in Fig. 2b (electroporation with a control shRNA) are indeed from a different experiment than those in Fig. 1d (electroporation with the fluorescent reporters only). We have removed the sentence from the legend, which now reads as follows: p29 line 738

"Scatter plots of Rmito in control cells (CTRL shRNA) and upon downregulation of CDKN1C (CDKN1C shRNA) at E3. Mann-Whitney. **: p=0.003".

•Fig. 2B. There is a double asterisk on the graph but only a single in the legend. Is the p value accurate?

The p value in the legend was incorrect. The correct value is $p=0.003$, which corresponds to a double asterisk. This error has been corrected (p29 line 739)

•Fig. 2D. In the lox mito-GFP image, the border of cell A drawn on the image excludes mitochondria included in the reconstruction on the right. Which is accurate?

As pointed out by Reviewer 1, the comparison of the two images in Fig. 2d can give the impression that a “mitochondrial patch” at the bottom of the image is wrongly included in cell A in the reconstruction. This impression arises from the fact that the axis of the two images is different: the confocal image is a true en-face image, while the reconstruction is slightly rotated around the y-axis as we felt that this angle better illustrates the difference of volume between A and B, also obvious in the animation in Supplementary Movie 3. To avoid confusing future readers, we have now replaced the 3D reconstruction with an image with the same axis of visualization as the confocal image. We have also better aligned the two images on the horizontal axis and precisely matched their scale (the reconstruction was slightly enlarged compared to the confocal image in the previous version of the figure). Additionally, we added an asterisk to the mitochondrial patch mentioned by the reviewer and noted in the legend of Fig. 2d that these mitochondria are not part of the sister cell pair (p30, line 748):

“The asterisk marks mitochondria belonging to a neighboring cell and which are not included in the 3D reconstruction of the sister cells pair.”

•Fig. 2E. These data are based on a good sample size of cells undergoing PN divisions; however, the number of PP and NN cell numbers a much lower. Given these differences in cell numbers, is it valid to compare differences in mean Rmito between these populations? The authors may want to consider increasing their sample sizes for this experiment.

We would like to clarify that we did not formally compare the means of Rmito in this dataset. Our analysis is only meant to highlight that the range of distribution of Rmito values among progenitors that divide symmetrically (PP and NN) is very different from the one of progenitors that divide asymmetrically (PN). In particular Rmito values below 0.85 are only observed in the PN population, with two exceptions in the PP group. The bars that were shown on the graph in Fig. 2e represented the median values. To avoid causing any potential confusion for readers, we have now removed them.

These results are reinforced by fate tracking analyses performed at E2, in which all but 3 out of 32 symmetrically dividing progenitors (31 PP and 1 NN) have a Rmito above 0.85 (Fig. 3g, left panel). This result appears later in the study and cannot be referred to when we describe the data of Fig. 2d. However, we have added a comment highlighting this observation in the legend of Fig. 3g: *“Note that similar to our observations at E3 (Fig. 2e), the majority of PP pairs in the control condition at E2 (28/31) show a Rmito>0.85.”*

•Fig. 2E. The authors should also define open and filled dots in 2E, although later defined in 2F.

We have now defined open and filled dots already in the legend of Fig. 2e as follows: p30 line 753: *“Open circles mark the only two cases of PN pairs with Rmito<0.85 where the largest mitochondrial pool is inherited by the future neuron.”* In addition, we removed the reference to Fig. 2e in the definition of filled versus open circles in the legend to Fig. 2f.

Results 3

•**Extended data Fig. 3. The authors may want to consider revisiting this data and providing cell numbers. In the representative images of neural tube sections provided, the difference in neuron numbers is not clear. In fact, there appear to be more HuC/D expressing cells on the electroporated sides of the control embryo compared to the CDC25B expressing cells. It would also be helpful to outline the region of transfected neurons to aid comparison of electroporated cells only.**

We have added the numbers of HuC/D and pRb cells that were counted in the legend of Supplementary Fig.3 and outlined the region of the mantle containing the electroporated neurons.

“CTRL shRNA: 1951 GFP positive cells from 7 embryos, CDC25B shRNA: 1875 GFP positive cells from 7 embryos”

The reviewer's observation that more cells express HuC/D on the electroporated side in the control is likely due to the slightly higher density of these double positive cells in the ventral area of this embryo.

We would like to clarify that our analyses selectively focused on the dorsal half of the neural tube (highlighted by a horizontal dotted white bar in the new version of the Supplementary Fig. 3). This *a priori* decision was dictated by the asynchrony of neurogenesis along the dorsal-ventral axis of the neural tube, where the ventral part differentiates earlier than the dorsal part. Specifically, neurogenesis has barely started in the dorsal area at the stage of our analysis, whereas most ventral progenitors are already neurogenic. We therefore expect to better highlight the role of CDC25B in inducing premature neurogenesis by focusing on the dorsal area, which was used for counting.

This methodological point was not included in the previous version of the manuscript. We have now corrected this oversight and added a paragraph describing this point in the Methods section as follows: p20 line 581 : **“We explored the effect of overexpressing CDC25B in progenitors on the production of neurons at the tissue level. In order to target and investigate specifically the neurogenic transition, we concentrated our analyses on the dorsal half of the neural tube where this transition has barely started at the time of analysis (E2.75, HH17) whereas most ventral progenitors are already neurogenic. Neuron and progenitor populations were evaluated at 30 hours after electroporation (hae) at E1.5 via immunofluorescence (GFP-pRb positive cells were considered as progenitors, GFP-HuC/D positive cells as neurons (Supplementary Fig. 3). Cell counts were performed on several distinct and non-overlapping z planes per vibratome section.”**

We have replaced the original image of the control embryo with an image from another vibratome section from the same embryo. The new image offers a better visual match for the comparison between the two conditions. In addition, we provide several new examples of higher magnifications in the dorsal region used for quantification to better illustrate the increase in GFP-electroporated HuCD-positive neurons.

The legend of the Supplementary Fig. 3 has been modified to incorporate all these changes.

“b. Top panels: representative transverse sections (6 μm z-projections) of the electroporated chick neural tube (thoracic level) at E2.75. Immunofluorescence with pRb antibody to label progenitors (red, nuclear signal) and with HuC/D antibody to label neurons (magenta, cytoplasmic signal) in control and overexpression of CDC25B (CDC25B OE) conditions. H2B-GFP fluorescence (green) represents the fluorescent electroporation reporter and DAPI staining is shown in cyan. The white horizontal dashed lines show the lower limit of the counting zone. The separation between ventricular and mantle zones in the counted region is marked by a yellow dashed line. Scale bar: 100 μm .

Bottom panels: representative close ups (4 μm z-projections) showing HuC/D positive electroporated cells (white arrows) and non-electroporated HuC/D positive cells (asterisks) in the control and overexpression conditions. Scale bar: 40 μm ».

Results 5

•The authors should expand upon the significance of CatchFire mediated changes in mitochondrial dispersion and basal localization and the rationale behind how this may lead to asymmetric inheritance of mitochondria.

Modifications of mitochondrial distribution in metaphase and anaphase, represent the first manifestation of the efficient tethering of mitochondria to the microtubule network. Interestingly, we did not observe cases displaying *unequal partitioning* without prior observation of changes of mitochondrial dispersion and basal localization in *metaphase-anaphase*, even in cells dividing at the beginning of the period of ligand exposure. In contrast, we did observe several cases towards the end of ligand exposure where the *modifications in meta/anaphase* were **not** followed by *unequal distribution*. This suggests that the artificial recruitment has to be built up during metaphase and anaphase AND maintained until cytokinesis to achieve an effective unequal distribution in the CatchFire condition. However, the number of divisions at the beginning and end of the ligand exposure window are too small in our sample to draw a definitive conclusion on whether the early modifications are a prerequisite for an asymmetric distribution of the mitochondrial pool.

During the process of revision, we realized that in our description of the two manifestations, the sentence p7line 154 :“Most importantly, **this led** to unequal mitochondrial segregation (...)” may have been misleading. The use of “this led” referred to the ligand administration, and was not meant to imply that the basal localization led to the unequal distribution. To avoid any confusion, we propose to reformulate the sentence as follows: “Most importantly, **we observed** unequal mitochondrial segregation (...)”.

The mechanism leading to the unequal segregation itself is still unclear. At least two possible interpretations can be put forward: in the first scenario, the unequal segregation is the consequence of the forced attachment to the microtubular network during mitosis, that will “cluster” them at the cleavage plane, resulting in a congestion that disrupts equal partitioning at the time of cytokinesis.

In the second scenario, the unequal mitochondrial transport is favored by an asymmetry of the microtubular network. This asymmetry could preexist, as has been shown for the transport of other cell determinants in other system (Derivery et al, doi: 10.1038/nature16443; Richard et al, 10.1038/s42003-024-06018-7), but not normally used at that stage by mitochondria for their mitotic segregation; it could also be created de novo (or amplified) by subtle modifications of the network by the CatchFire system itself, and linked to the forced tethering of mitochondria. The existence of such an asymmetry represents a promising research direction for future studies on the endogenous mechanism of unequal mitochondrial distribution in asymmetric divisions.

To expand on this, as well as to incorporate data relative to the state of the microtubule network (in response to this reviewer’s next question), we propose to add a comment after the first description of the CatchFire system’s effects on mitochondrial distribution: (p 7, line 155):

“The mechanism leading to the unequal segregation itself is unclear. It may simply be the result of the clustering of mitochondria creating a congestion near the cleavage plane that would in turn perturb their equal partitioning at the time of cytokinesis. Alternatively, the artificial tethering of mitochondria

to microtubules may reveal or create by itself an asymmetry in the microtubule network, resulting in their directional transport. A gross disruption of the microtubule network that would perturb the normal mechanism of equal mitotic mitochondrial partitioning at E2 is not supported by experiments showing no visible alterations of the network integrity in mitosis in the CatchFire condition (Supplementary Fig. 5b). »

•Is it possible that CatchFire could have caused ACD because of disruption to microtubules rather than mitochondria? This possibility should at least be discussed.

We have now performed additional experiments to explore the integrity of the microtubule network in CatchFire conditions. After labelling microtubules with SiR-tubulin, we compared the microtubule network in Catchfire and control conditions at shorter time intervals and focused on the last steps of mitosis. We did not observe any obvious differences between the two conditions. To illustrate this new experiment, sequential images of SiR-tubulin labelling immediately before and after cytokinesis have now been added in a new panel b to the ones already shown in Supplementary Fig. 5a. These experimental results render the scenario in which a major disruption of the microtubule network causes ACD unlikely. As indicated above, this experiment is now mentioned in the text (p. 7, line 160):

“A gross disruption of the microtubule network that would perturb the normal mechanism of equal mitotic mitochondrial partitioning at E2 is not supported by experiments showing no visible alterations of the network integrity in mitosis in the CatchFire condition (Supplementary Fig. 5b). »

More subtle modifications in the organization of the network, such as the creation (or amplification) of an asymmetry could be missed with the current resolution of our SiR-tubulin images. Exploring such subtle effects falls within the scope of a detailed characterization of the endogenous mechanism of unequal segregation of mitochondria during neurogenesis, which represents a logical and exciting follow-up to the present work. Indeed, a central spindle asymmetry has already been shown to bias the transport of other organelles in other models (Derivery et al, doi: 10.1038/nature16443; Richard et al, 10.1038/s42003-024-06018-7).

Notably, in the absence of ligand, we did not observe any fate changes. This excludes the possibility that the expression of the CatchFire components, and in particular of the Kif17 motor domain fused to the FireTag, which interacts with the microtubule network, leads on its own to ACD. Thus, the tethering and unequal distribution of mitochondria is mandatory to induce asymmetric fates in this experimental scheme.

Discussion

**•Discussion should be expanded with more critical interpretation of their findings.
•Line 218. They argue against qualitative differences in mitochondria affecting cell fate, suggesting that volume is important, but both could be true.**

In response to this request, we have reformulated and slightly expanded the discussion to offer a more nuanced interpretation of the findings, and of the “opposition” between quantitative and qualitative parameters. The new paragraph reads as follows: p9 line 214:

“Our experimental manipulation of mitochondrial distribution during mitosis combined with fate tracking in vertebrate neural progenitors in vivo represent the first direct causal demonstration that mitochondria act as asymmetric cell fate determinants. Previous work in several models of asymmetric division has correlated the description of an asymmetric inheritance of mitochondria with distinct

morphologies or functions during mitosis to functional data showing their involvement in divergent fates in the progeny³⁻⁸. These studies have pointed to a role of differential qualitative characteristics of the inherited mitochondrial pools in cell fate acquisition. ~~However, our results with the CatchFire system, showing that an experimentally induced imbalance in the inheritance of mitochondrial volume during the progenitor amplification phase is sufficient to induce neural differentiation, tend to argue against it in our model.~~ By contrast, our work identifies the importance of inheriting quantitatively unequal mitochondrial pools in the establishment of a different fate between sister cells. In particular, the CatchFire experiments show that an experimentally induced imbalance in the inheritance of mitochondrial volume drives neural differentiation during the progenitor amplification phase. While our results place the quantitative parameter at the heart of the asymmetric division mechanism, additional qualitative parameters within the inherited mitochondrial pools may also contribute to the final fate decision. In particular, this might be the case as more committed progenitors progressively acquire mitochondrial functional characteristics associated with differentiation²⁵⁻²⁶. These parameters could play a key role in the subset of PN pairs in our study where the inheritance was quantitatively balanced (Fig. 2e)."

In addition, we have modified and expanded the last paragraph of the discussion in response to a comment from Reviewer 3: (p11, line 263)

"Our *in vivo* data establish a causal link between a relative imbalance in the inheritance of mitochondria and differential fate acquisition between sister cells. ~~Future studies will be needed to decipher how unequal mitochondrial segregation is achieved, and whether it is coordinated with the mitotic distribution of other asymmetric determinants to robustly specify fate decisions during neurogenesis.~~ Previous studies have proposed other determinants of identity during vertebrate neurogenesis, amongst which several regulators of the Notch signaling pathway^{18,30-34}. As shown here for mitochondrial segregation, the unequal partitioning of each of these determinants shows a strong, but imperfect correlation with divergent fate between sister cells when characterized with a cellular resolution. Future studies will be needed to decipher whether they contribute synergistically to robust cell fate choices.

How is unequal mitochondrial segregation achieved? One attractive hypothesis is that they use an asymmetry of the central spindle for their routing towards the end of mitosis, as described for the unequal partitioning of SARA-endosomes in drosophila and zebrafish neural progenitors^{31,35}. More generally, this could represent a shared mechanism coordinating the asymmetric transport of different organelles involved in fate decisions³⁰⁻³⁴. Finally, deciphering how multiple determinants using different routes for their unequal distribution are coordinated represents a promising avenue for understanding how cellular processes integrate into developmental decisions^{18,30-33}."

Reviewer #2 (Remarks to the Author): *I co-reviewed this manuscript with one of the reviewers who provided the listed reports. This is part of the Nature Communications initiative to facilitate training in peer review and to provide appropriate recognition for Early Career Researchers who co-review manuscripts.*

Reviewer #3 (Remarks to the authors)

Bunel et al.

This is an excellent and very well written manuscript that will have a major impact on our understanding of the role of mitochondria in cell fate decisions in the context of asymmetric cell division. The authors used state of the art technology, the experiments were done in a very rigorous way, and the data is convincing. Overall, I very much enjoyed reading it. This is great work, and the authors can be congratulated.

Below are a few comments and questions.

1. In Fig S2, the authors present data that R_{mito} and R_{cell} do not correlate. However, it is unclear why they did not consider cell volume in all of their data by determining mitochondrial volume per cell volume (mito/cell) for each daughter cell and then taking the ratio of that. I can see that both is of value, but it would be interesting to see how the data looks then, and it might hint at an interaction between mitochondria and the nucleus/genome or the cytoplasm/other organelles.

In response to the reviewer's point, we have measured both the cell volume and the mitochondrial volume for each daughter cell for 52 of the 58 tracked Tis-Cre pairs (and in a subset of 18 Control PP pairs at E2), and recalculated R_{mito} after normalizing the mitochondrial volume of sister cells to the total cell volume (nR_{mito}).

Using the same 0.85 threshold to define equal versus unequal inheritance, the distribution of nR_{mito} did not change in a major way compared to R_{mito} within the symmetrically dividing progenitor population in the Tis21-Cre cohort: within the PP population, one pair switched from equal to unequal and another from unequal to equal, leaving the number of pairs with $nR_{mito} > 0.85$ unchanged. Within the NN population, one pair switched from equal to unequal. Similar results were obtained in PP pairs at E2, with one out of 18 PP pairs switching from unequal to equal. Changes are more pronounced within the PN population in the Tis-Cre dataset: seven pairs switch from unequal to equal while three pairs switch from equal to unequal.

Overall, this normalized ratio produces slightly different results compared to R_{mito} ; the ratio of the proportion of the cellular volume occupied by the mitochondria (nR_{mito}) seems to be a less accurate predictor of an asymmetric fate than the original ratio of the quantity of mitochondria (R_{mito}). Therefore, we think that using the absolute mitochondrial volume rather than normalized to the cell volume is a better metric in this study.

The following graph shows the matched R_{mito} and normalized R_{mito} values for the 52 Tis21-Cre pairs (due to a weak reporter expression, the cellular volume could not be accurately measured in 4PP and 2PN pairs of the 58 pairs of this dataset) and 18 E2 PP pairs. For each condition, pairs that show a large change between R_{mito} and nR_{mito} are shown as black dots. Of note, several pairs that switch category but show only a minor difference near the threshold (between 0.84 and 0.86) are not highlighted.

.....

2. Along those lines, have the authors looked at the inheritance of other organelles in their system?

We have now extended our research to include the study of ER distribution in a population of neural progenitor daughter cells at E3, employing a methodology similar to the one used for the investigation of mitochondrial segregation.

Using live imaging, we studied the partitioning of this network, marked by a fluorescent construct (KDEL-GFP) (PMID: 26383951), and measured the ratio (R_{ER}) of its volume between sister cells at E3, when the distribution of R_{mito} is the most heterogeneous. Our results show that the distribution of R_{ER} values differs from the distribution of R_{mito} . R_{ER} distribution is much more homogeneous with the vast majority of the values falling within the 0.8-1 range. These results suggest that organelle segregation asymmetry is not a universal feature of asymmetric division.

The data are shown below. Left panel: graph showing the distribution of values of 42 KDEL-GFP cells, and R_{mito} values for 33 of these cells. Right panel: two representative examples showing a homogenous distribution of the ER in cases of equal (top) and unequal (bottom) mitochondrial inheritance.

A paired student's t-test (*, $p=0.0189$) comparing the R_{ER} and R_{mito} values of the 33 pairs (blue color) for which we measured both parameters indicates that the two distributions are different.

Concerning the general question of organelles, we now mention more explicitly in the ‘Discussion’ section several studies describing the unequal partitioning of other organelles, and whether their transport could be coordinated and their activity synergize with mitochondria to secure cell fate decisions: p11, line273

“How is unequal mitochondrial segregation achieved? One attractive hypothesis is that they use an asymmetry of the central spindle for their routing towards the end of mitosis, as described for the unequal partitioning of SARA-endosomes in drosophila and zebrafish neural progenitors^{31,35}. More generally, this could represent a shared mechanism coordinating the asymmetric transport of different organelles involved in fate decisions³⁰⁻³⁴. Finally, deciphering how multiple determinants using different routes for their unequal distribution are coordinated represents a promising avenue for understanding how cellular processes integrate into developmental decisions^{18,30-33}.”

3. In Fig 1C, it appears that mitochondria in the ‘even’ example are distributed throughout the daughter cells whereas in the ‘uneven’ example, they are more organized, clustered around what was the division plane. In Fig S1 it seems the opposite, the uneven example seems more distributed throughout the cell and the even example more clustered. (In the CatchFire experiment and Fig S6, similar phenomena are observed and referred to as ‘distribution’.) Can the authors elaborate on this? Could it be slightly different timepoints after cytokinesis? Or do the authors think there is control of distribution prior to cell division? As it stands, this is confusing.

Concerning the question of **the control of distribution prior to cell division**: we had already investigated in a few cases of progenitor divisions whether the unequal mitochondrial segregation observed in their progeny was predictable before mitosis. We monitored the distribution of mitochondria with respect to the chromosomal plate in the mother cell at several time points before mitosis (each 3 minutes between -15 minutes and cytokinesis). We found that the mitochondrial distribution is very labile during mitosis, probably due in part to the persistent rotation of the mitotic spindle in neural progenitors. The future distribution of mitochondria after mitosis (unequal or equal) becomes predictable only one time point (3 minutes) before cell division. This observation, made in a subset of our data, does not favor the hypothesis that mitochondrial distribution is controlled prior to cell division.

Concerning the morphology and density of the mitochondrial network within the daughter cells: as pointed by the reviewer, they appear slightly variable between the examples presented in our figures. As suggested, differences in post-cytokinesis timing may be responsible for these differences, since the cell reorganizes very quickly after cytokinesis and the space available for all organelles is remodeled. We explored this by acquiring full 3D stacks at 1min intervals during the first time points after mitosis. These sequential images within single pairs reveal modest changes of the network in this time window, that indeed resemble the differences that we observe between different pairs (four examples are shown in the figure below). Another very likely possibility is that these differences result from differences in the cleavage plane of the division and/or imperfections in the montage (i.e; the montage is not perfectly flat), since looking at 3D reconstructions through different angles also modifies the visual impression of the network organization.

To further investigate this question, we evaluated the dispersion of the mitochondrial network in the tracked Tis21-Cre cell population using the D_i metric described in the CatchFire experiments (Supplementary Figure 6) as a proxy for this organization. This measurement shows modest variations in D_i within this population ($0.61 < D_i < 0.81$ —the same range as seen at E2 in the CatchFire controls at E2). These variations are not correlated with R_{mito} , and as a consequence they are not correlated with fate (PP, PN or NN). For all these reasons, we believe that the distribution/dispersion of the network (measured as we have done here) is unlikely to be involved in asymmetric inheritance/fate.

The left panel below shows the distribution of R_{mito} and D_i in the Tis21-Cre population. The right panel reproduces the graph from Supplementary Figure 6, showing the data for control and CatchFire cells at E2.

The situation is different in the context of the CatchFire experiments in presence of the ligand, in which experimentally induced mitochondrial compaction correlates with an ‘unequal’ R_{mito} and with asymmetric fate. It should nevertheless be noted that the compaction observed in this situation is not comparable with the one discussed above, as it is much greater than the one observed during neurogenesis ($D_i < 0.6$, green dots in the graph above).

4. Cell fate tracking and CatchFire experiment. In the abstract the authors state “We set up a chemogenetic approach to experimentally displace mitochondria specifically during mitosis to force their unequal inheritance in vivo and we found that this was sufficient to drive premature neuronal differentiation.” However, they are more careful in the title, which is “Unequal mitochondrial segregation promotes asymmetric fates during neurogenesis”. Data shown in Fig 2D, Fig. 3G and Fig S8C, suggest that low R_{mito} (below 0.85) is neither required nor sufficient for asymmetric fates (P/N) (for example R_{mito} of 0.65 in PP divisions and 32% of P/N divisions have R_{mito} of 1-0.85). I agree with the authors that unequal mitochondrial segregation promotes P/N divisions but I think they need to be a bit more careful when using the term ‘sufficient’. Based on their data, other cell fate determinants are likely to play a role and contribute, and this should be stated and discussed.

Based on the reviewer’s comment, we have reworded the sentence in the abstract as follows: p2

“We set up a chemogenetic approach to experimentally displace mitochondria specifically during mitosis to force their unequal inheritance in vivo and we found that **this was sufficient to drives** premature neuronal differentiation.”

Furthermore, as the reviewer points out, acquiring a secure fate is likely to be multifactorial, and other determinants of identity have been identified during neurogenesis. Each of them shows a strong, but imperfect, correlation with divergent fates between sister cells, suggesting that they could contribute synergistically to the robustness of binary decision-making. Understanding how/whether the segregation of these determinants is coordinated to secure fate represents a promising avenue for this project.

The new formulation of the end of the discussion also takes this point into account: p11 line267

« Previous studies have proposed other determinants of identity during vertebrate neurogenesis, amongst which several regulators of the Notch signaling pathway^{18,30-34}. As shown here for mitochondrial segregation, the unequal partitioning of each of these determinants shows a strong, but imperfect correlation with divergent fate between sister cells when characterized with a cellular resolution. Future studies will be needed to decipher whether they contribute synergistically to robust cell fate choices.

How is unequal mitochondrial segregation achieved? One attractive hypothesis is that they use an asymmetry of the central spindle for their routing towards the end of mitosis, as described for the unequal partitioning of SARA-endosomes in drosophila and zebrafish neural progenitors^{31,35}. More generally, this could represent a shared mechanism coordinating the asymmetric transport of different organelles involved in fate decisions³⁰⁻³⁴. Finally, deciphering how multiple determinants using different routes for their unequal distribution are coordinated represents a promising avenue for understanding how cellular processes integrate into developmental decisions^{18,30-33}.”

5. Figure 3 Parts C and D. I found this not easy to follow. Maybe those schematics could be simplified or the labelling changed.

To simplify Figure 3C, we have now removed the “daughter fate assignment using long-term live imaging tracking”, since it is already described in Figure 2C, and only describe the outline of the protocol for shorter tracking followed by fate assignment via immunolabeling, as it corresponds to the detailed example presented in figure 3D. In addition, we have color-coded the images corresponding to z-levels of the two sister cells in order to better illustrate the correspondence between the schematics of the time course and the confocal images. The same modifications have been added in the extended data figure 7 that shows examples of PP and NN pairs identified in short term tracking + immunolabeling.

We have reworded the legend in Fig. 3 as follows: p30 line 764

“(c) Scheme of combined live monitoring of mitochondrial inheritance and daughter cell fate allocation by immunolabeling at E2.5.

(d) Scheme (top) and en-face views (middle) of daughter cell’s nucleus (H2B-GFP, white) in the depth of the neuroepithelium during the live tracking period (left) and correspondence with pRb immunofluorescence at its end (right, magenta). Images corresponding to z-levels of the two sister cells are color coded (dotted frame) to illustrate the correspondence with the schematics of the time course. Bottom: $R_{\text{mito}}=0.78$ in the mother cell matches with PN fate: Scale bar: 5 μm . Asterisks: neighboring cells used for registration between live and fixed images.”

and the legend in Supplementary Fig. 7 as follows: *“Images corresponding to z-levels of the two sister cells are color coded (dotted frame) to illustrate the correspondence with the schematics of the time course.”*

6. The authors state that their data indicates that mitochondrial volume (quantity) rather than a qualitative aspect is important for asymmetric fate. However, they do not provide any evidence that there is no qualitative difference between mitochondria in P/N sisters.

The submitted version of the manuscript “opposed” our quantitative observations to previous studies highlighting qualitative aspects. In response to a request by reviewer 1 on the same topic, we have

reformulated and slightly expanded the discussion to offer a more nuanced interpretation of the findings. The new paragraph reads as follows (p9, line 214):

“Our experimental manipulation of mitochondrial distribution during mitosis combined with fate tracking in vertebrate neural progenitors in vivo represents the first direct causal demonstration that mitochondria act as asymmetric cell fate determinants. Previous work in several models of asymmetric division has correlated the description of an asymmetric inheritance of mitochondria with distinct morphologies or functions during mitosis to functional data showing their involvement in divergent fates in the progeny³⁻⁸. These studies have pointed to a role of differential qualitative characteristics of the inherited mitochondrial pools in cell fate acquisition. ~~However, our results with the CatchFire system, showing that an experimentally induced imbalance in the inheritance of mitochondrial volume during the progenitor amplification phase is sufficient to induce neural differentiation, tend to argue against it in our model.~~ By contrast, our work identifies the importance of inheriting quantitatively unequal mitochondrial pools in the establishment of a different fate between sister cells. In particular, the CatchFire experiments show that an experimentally induced imbalance in the inheritance of mitochondrial volume drives neural differentiation during the progenitor amplification phase. While our results place the quantitative parameter at the heart of the asymmetric division mechanism, additional qualitative parameters within the inherited mitochondrial pools may also contribute to the final fate decision. In particular, this might be the case as more committed progenitors progressively acquire mitochondrial functional characteristics associated with differentiation^{25,26}. These qualitative parameters could play a key role in the subset of PN pairs in our study where the inheritance was quantitatively balanced (Fig. 2e).”

Have the authors analyzed for example mitochondrial morphology?

Following the previous point (quantitative versus qualitative), differences in the morphology of the inherited pools is indeed one of the parameters that could influence cells fate. The fast spinning disk imaging used in our study was optimal to obtain temporal and 3D dynamics in the tissue in many different experimental conditions and to catch the precise timing of cytokinesis at which we performed our analysis. Nevertheless, it does not provide sufficient resolution to identify subtle morphological differences in the network in daughter cells immediately after cytokinesis. This has so far precluded the exploration of this parameter in our dataset.

Investigating qualitative parameters, and in particular the morphology of this highly dynamic organelle, and match them with fate decisions in the progeny via direct live tracking, presents important challenges, especially in vivo, but will be important future directions.

7. Along these lines, the CatchFire system could potentially bias towards the tethering of small organelles. Have the authors seen any evidence of that?

We have not specifically monitored any other organelles in the CatchFire condition. However, we would like to clarify that the Kif17-FireMate construct is a truncated version of Kif17 (aa 1-547 / 1028) that essentially contains the motor domain of kif17 (4-335) and should therefore not interact with cellular cargoes. In particular, it should not tether other cellular organelles, which lack the CatchFire Tag. This information has now been added in the manuscript (p.7 line 149):

*“We electroporated vectors expressing fusions of the CatchFire dimer moieties, one targeted to the outer mitochondrial surface (MitoFireMate) and the other fused to a kinesin **motor domain** (KifFireTag).....”*

In addition, if kif17 expression by itself induced a transport bias in other organelles, it would be expected to happen in the absence of the ligand. If such a transport exists, it does not affect fate, since in the control cells electroporated with the CatchFire constructs, we observe a majority of PP divisions, as expected at this stage.

And could CatchFire impact the cell cycle by activating an organelle inheritance check point? Have the authors observed cell cycle delays?

This is an interesting suggestion, as it has been proposed that the correct partitioning of several organelles is required for correct mitotic progression and/or spindle formation (reviewed in Mascanzoni et al, DOI: 10.3389/fcell.2019.00133). As shown in our Supplementary Fig. 5, the mitotic spindle itself, based on SiR-tubulin staining, is formed and appears normal in our CatchFire experiments. Besides, in these experiments, we did not observe any mitotic arrest that could indicate failure to pass such a checkpoint. In addition, we did not observe any differences in the duration of mitotic phases (from metaphase to cytokinesis) between cells dividing before and after addition of the ligand. Overall, this indicates that the CatchFire scheme does not affect mitotic progression. More generally, the duration of the CatchFire treatment is short, and it is not expected to impact the cell cycle as a whole.